# Measurement-based climatology of aerosol direct radiative effect, its sensitivities, and uncertainties from a background southeast U.S. site

James P. Sherman[1] and Allison McComiskey[2]

[1]Department of Physics and Astronomy, Appalachian State University, 525 Rivers Street, Garwood Hall
room 231, Boone, NC 28608 USA
[2]NOAA Earth Systems Research Laboratory, Global Monitoring Division / GMD-1, 325 Broadway,
Boulder, CO 80305 USA

*Correspondence to*: James P. Sherman (shermanjp@appstate.edu)

**Abstract**

Aerosol optical properties measured at Appalachian State University's co-located NASA AERONET and NOAA ESRL aerosol network monitoring sites over a nearly four-year period (June 2012 thru Feb 2016) are used, along with satellite-based surface reflectance measurements, to study the seasonal variability of diurnally averaged clear sky aerosol direct radiative effect (DRE) and radiative efficiency (RE) at the top-of-atmosphere (TOA) and at the surface. Aerosol chemistry and loading at the Appalachian State site are likely representative of the background southeast U.S. (SE U.S.), home to high summertime aerosol loading and one of only a few regions not to have warmed during the 20[th] century. This study is the first multi-year 'ground truth' DRE study in the SE U.S., using aerosol network data products that are often used to validate satellite-based aerosol retrievals. The study is also the first in the SE U.S. to quantify DRE uncertainties and sensitivities to aerosol optical properties and surface reflectance, including their seasonal dependence.

Median DRE for the study period is -2.9 Wm$^{-2}$ at the TOA and -6.1 Wm$^{-2}$ at the surface. Monthly median and monthly mean DRE at the TOA (surface) are -1 to -2 Wm$^{-2}$ (-2 to -3 Wm$^{-2}$) during winter months and -5 to -6 Wm$^{-2}$ (-10 Wm$^{-2}$) during summer months. The DRE cycles follow the annual cycle of aerosol optical depth (AOD), which is 9 to 10 times larger in summer than in winter. Aerosol RE is anti-correlated with DRE, with winter values 1.5 to 2 times more negative than summer values. Due to the large seasonal dependence of aerosol DRE and RE, we quantify the sensitivity of DRE to aerosol optical properties and surface reflectance, using a calendar day representative of each season (DEC 21 for winter; MAR 21 for spring, JUN 21 for summer, and SEP 21 for fall). We use these sensitivities along with measurement uncertainties of aerosol optical properties and surface reflectance to calculate DRE uncertainties. We also estimate uncertainty in calculated diurnally-averaged DRE due to diurnal aerosol variability. Aerosol DRE at both the TOA and surface is most sensitive to changes in AOD, followed by single-scattering albedo ($\omega_0$). One exception is under the high summertime aerosol loading conditions (AOD$\geq$0.15 at 550 nm), when sensitivity of TOA DRE to $\omega_0$ is comparable to that of AOD. Aerosol DRE is less sensitive to changes in scattering asymmetry parameter (g) and surface reflectance (R). While DRE sensitivity to AOD varies by only ~25 to 30 % with season, DRE sensitivity to $\omega_0$, g, and R largely follow the annual AOD cycle at APP, varying by factors of 8 to 15 with season. Since the measurement

uncertainties of AOD, $\omega_0$, g, and R are comparable at Appalachian State, their relative contributions to DRE uncertainty are largely influenced by their (seasonally dependent) DRE sensitivity values, which suggests that the seasonal dependence of DRE uncertainty must be accounted for. Clear sky aerosol DRE uncertainty at the TOA (surface) due to measurement uncertainties ranges from 0.45 $Wm^{-2}$ (0.75 $Wm^{-2}$)

for DEC to 1.1 $Wm^{-2}$ (1.6 $Wm^{-2}$) for JUN. Expressed as a fraction of DRE computed using monthly median aerosol optical properties and surface reflectance, the DRE uncertainties at TOA (surface) are 20 to 24 % (15 to 22 %) for MAR, JUN, and SEP and 49 % (50 %) for DEC. The relatively low DRE uncertainties are largely due to the low uncertainty in AOD measured by AERONET. Use of satellite-based AOD measurements by MODIS in the DRE calculations increases DRE uncertainties by a factor

of 2 to 5 and DRE uncertainties are dominated by AOD uncertainty for all seasons. Diurnal variability in AOD (and to a lesser extent g) contributes to uncertainties in DRE calculated using daily-averaged aerosol optical properties that are slightly larger (by ~20 to 30%) than DRE uncertainties due to measurement uncertainties during summer and fall, with comparable uncertainties during winter and spring.

## 15   1        Introduction

Predictions of future climate change resulting from projected increases in carbon dioxide are limited in part by uncertainties in the direct and indirect radiative forcing due to aerosols (Andreae, et.al, 2005). On a global average, the measurement-based estimates of aerosol direct radiative effect (DRE) are 55 to 80 % greater than the model-based estimates. The differences are even larger on regional scales and for

the anthropogenic component (Yu, et.al, 2006). Such measurement-model differences are a combination of differences in aerosol amount (aerosol optical depth-AOD), single-scattering properties, surface albedo, and radiative transfer schemes (Yu et al., 2006). As part of a radiative transfer closure study, Michalsky et al.(2006) found that six radiative transfer models (RTMs) were all able to simulate clear-sky direct and diffuse shortwave fluxes to within 1.0% and 1.9%, respectively, of the measured fluxes,

provided that all models used the same co-located measurements of the aerosol optical properties. They concluded that the largest source of difference in the RTM-calculated fluxes is likely due to how the RTM extrapolates the aerosol optical properties used as inputs (particularly AOD) to unspecified wavelengths. As a follow-up to this study, McComiskey et al. (2008) showed that the sensitivities of clear-sky DRE to

changes in aerosol inputs were not dependent on the model used. Both studies demonstrate that the RTMs are capable of calculating clear-sky DRE with high precision and that DRE uncertainty arises largely from incorrectly-specified aerosol optical properties, which can result from lack of regionally-representative values, measurement uncertainties, and spatio-temporal aerosol variability. One of the high-priority tasks recommended (Remer, et.al 2009) to reduce the uncertainty in aerosol radiative effects is to "*Maintain, enhance, and expand the surface observation networks measuring aerosol optical properties for satellite retrieval validation, model evaluation, and climate change assessments.*"

The southeast U.S. (SE U.S.) is home to some of the highest warm-season aerosol loading in the U.S. (Goldstein et al., 2009) and is also one of only a few regions not to have not exhibited a warming trend in the 20th century (Menne et al., 2009). Several studies conducted during the past two decades have attempted to quantify aerosol DRE in the SE U.S. Yu et al. (2001) applied 34 days of aerosol optical property measurements near Mount Mitchell, NC from June thru December 1995 to estimate variability in SE U.S. aerosol DRE and atmospheric absorption by aerosols, along with an estimate of 'annually averaged' DRE. Carrico et al. (2003) applied measurements of AOD and other aerosol optical properties as part of the Atlanta Supersite 1999 study to estimate summer top-of-atmosphere (TOA) DRE in urban Atlanta, GA. Goldstein et al. (2009) used region and time-averaged AOD near 550 nm, measured by the Multi-angle Imaging Spectrometer (MISR) aboard the polar-orbiting Terra satellite from 2000 thru 2007, as inputs to a first-order radiative transfer calculation (Haywood and Shine, 1995) to show that high summer AOD in the SE U.S. led to more negative aerosol TOA DRE in summer than winter (by 3.9 Wm$^{-2}$). Goldstein et al. (2009) hypothesized that this summer regional cooling effect was dominated by secondary organic aerosols, resulting from the oxidation of biogenic volatile organic compounds in the presence of anthropogenic NOx and SO$_2$. However, their DRE calculation used assumed values (rather than measured) for surface reflectance and for all aerosol properties except AOD, and they did not consider seasonal variations in these properties. Alston and Sokolik (2016) applied 12 years (2000 thru 2011) of AOD at 550 nm, cloud fraction, and surface albedo measured by the Moderate Resolution Imaging Spectrometer (MODIS) aboard Terra, along with single-scattering albedo near 550 nm from MISR, as inputs to the same TOA DRE equation (Eq. (2) of Haywood and Shine, 1995) used by Goldstein et al. (2009). Their primary objectives were to study TOA DRE seasonal variability and long-term trends

in the SE U.S., in the context of changes in AOD, cloud fraction, and surface albedo. They concluded that AOD was a major driver of regional TOA DRE (as compared to surface albedo and cloud fraction) and they also reported a decreasing linear trend in MODIS Terra AOD, which contributed to a small increasing trend (i.e. less negative) in TOA aerosol DRE. However, the sensitivities of DRE to aerosol single-scattering properties and surface reflectance were not explicitly quantified, nor were DRE uncertainties. Estimates of aerosol DRE using MODIS-measured AOD have higher uncertainties than those based on spectral AOD measured at NASA Aerosol Robotic Network (AERONET; Holben et al., 1998) sites, as discussed in Sect. 5.3 of this paper. The MODIS Collection 5.1 AOD also has been found to possess a consistently negative bias (0.02 to 0.03) over the rural SE U.S. Appalachian State AERONET site (Sherman et al., 2016a), which could lead to an under-estimation of aerosol DRE (i.e. less negative).

Ground-based sites as part of the NOAA Earth Systems Research Laboratory (NOAA ESRL; Delene and Ogren, 2002), NASA AERONET, and NASA Micro-pulsed Lidar (NASA MPLNET; Welton et al., 2001) federated aerosol monitoring networks possess continuous long-term records of aerosol optical properties used to evaluate aerosol DRE (McComiskey et al., 2008; Michalsky, et al., 2006). Established in 2009, the Appalachian Atmospheric Interdisciplinary Research Facility at Appalachian State University is home to the only co-located NOAA ESRL, NASA AERONET, and (since 2016) NASA MPLNET sites in the SE U.S. Aerosol chemistry and loading at the semi-rural, high-elevation Appalachian State site (referred to as APP in this paper) are likely representative of the background SE U.S. (Link et al., 2015). As such, APP is well-positioned to improve understanding of aerosol DRE in the SE U.S., including seasonal DRE variability, sensitivities, and uncertainties, as recommended by Remer et al. (2009).

The objective of this paper is to complement previous studies of aerosol DRE in the SE U.S. through a detailed, multi-year study of aerosol DRE seasonal variability, sensitivities, and uncertainties from a single ground-based aerosol network site. Specifically, we

1. Quantify the seasonal variability in diurnally averaged, clear sky aerosol DRE and direct radiative efficiency (RE=DRE per unit AOD) at APP, both at the TOA and at the surface, along with seasonal variability in aerosol and surface properties influencing DRE (Sect. 5.1)

2. Quantify the sensitivity of DRE to key aerosol and surface properties, including any seasonal dependence (Sect. 5.2)

3. Apply the DRE sensitivities (2) to calculate the uncertainty in DRE due to measurement uncertainties and due aerosol diurnal variability (Sects. 5.3 and 5.4)

Daily averaged aerosol optical properties measured on 418 days between June 2012 and February 2016 are used along with monthly averaged spectral surface reflectance measured by MODIS to study the annual DRE and RE cycles at APP (Sect. 5.1). In Sect. 5.2, we follow a similar approach to that used by McComiskey et al (2008) to quantify DRE sensitivity to AOD, single-scattering albedo ($\omega_0$), scattering asymmetry parameter (g), and surface reflectance (R). The DRE sensitivities are then used along with measurement uncertainties in AOD, $\omega_0$, g, and R to estimate the resulting uncertainties in DRE at the TOA and at the surface (Sect. 5.3). We then estimate diurnal variability in AOD, $\omega_0$, g at APP and use these, along with the DRE sensitivities, to estimate uncertainties in calculated diurnally-averaged DRE due to the use of daily-averaged aerosol inputs by the RTM (Sect.5.4). The use of well-established measurement protocols developed by NOAA ESRL (Delene and Ogren, 2002) and NASA AERONET (Holben et al., 1998), and possessing known uncertainties (Eck et al., 1999; Sherman et al., 2015), facilitates the first study of DRE sensitivities and measurement uncertainty in the SE U.S., with results that are directly comparable with other regions.

This paper differs from the aerosol DRE sensitivity and uncertainty analysis conducted by McComiskey et al. (2008) in that it addresses a different geographic region and in the following additional ways:

1. McComiskey et al (2008) considered generally representative properties of three surface aerosol sites and no seasonal dependence of DRE, while this study uses direct measurements and focuses on the seasonal DRE dependence at a single site

2. The use of measured values for all aerosol properties allows for us to consider their covariances in the DRE uncertainty calculations

3. We compare DRE uncertainties using ground-based AOD measurements made as part of AERONET with those using satellite-based AOD measurements from MODIS

4. We compare DRE uncertainties due to aerosol measurement uncertainty with those due to diurnal aerosol variability

For clarity, it is important to distinguish between aerosol DRE and the often-referenced aerosol direct radiative forcing (DRF), in addition to defining 'clear sky' DRE. Direct radiative effect refers to the difference in net radiative fluxes (Eq. (5)) at a given atmospheric level (often the TOA or surface) with and without the presence of atmospheric aerosols, while DRF refers to the anthropogenic component (Kaufman et al., 2005). Clear sky DRE refers to DRE calculated assuming cloud-free conditions, which amounts to turning clouds off in the RTM used to calculate the radiative fluxes. Most studies neglect cloud effects not only for simplicity but also because satellite-based aerosol retrievals can only be made in the absence of clouds. First-order DRE calculations such as those provided in Haywood and Shine (1995) account for aerosol DRE in the presence of clouds by multiplying the clear sky DRE by the cloud-free sky fraction. For clarity, we also include a table of commonly used acronyms and symbols used in this paper (Table A1 of Appendix A).

## 2    Site description

The APP site is located at the highest point on the Appalachian State University campus in Boone, NC (Fig.1). Lower tropospheric aerosols are sampled from a 34 m tower as part of NOAA ESRL, from which aerosol optical and microphysical properties (Sect. 3.1.2) are measured by in situ instruments (Sherman et al., 2015). Vertical profiles of aerosols and clouds have also been measured continuously by a micro-pulsed lidar (MPL) as part of NASA MPLNET (Welton et al.,2001) since March 2016. Lidar-measured vertical profiles of normalized relative aerosol backscatter were made periodically from 2011 thru 2014 (prior to joining MPLNET) but have no quality assurance and therefore are not used in this paper, other than qualitative inspection to verify that aerosols are largely confined to the lowest 1 to 2 km of atmosphere above APP. The region surrounding APP is heavily forested and possesses a diversity of elevations (< 300 m to > 2000 m) and a variety of weather regimes (i.e.., winter storms, convective cells, dying tropical cyclones, and stagnant summertime episodes). The region also includes a diversity of anthropogenic and biogenic aerosol sources. Lower tropospheric aerosol light scattering and absorption coefficients measured at APP are dominated by particles with diameter less than 1 μm (Sherman et al.,

2015) and sub-1 μm aerosol mass consists primarily of organics, with lower levels of sulfates (supplement to Link et al., 2015). Summer AOD in the SE U.S. (including APP) is influenced by isoprene-derived secondary organic aerosol (Goldstein *et al.*, 2009; Link et al., 2015). A biomass burning influence is present in winter aerosol mass concentrations measured at APP (supplement to Link *et al.*, 2015), likely due to residential wood burning in the region. Wood-burning stoves serve as the primary heating source for 6.2 % of occupied housing units in Watauga County (U.S. Census Bureau, 2010) and likely a larger percentage of housing units in the surrounding rural mountain communities.

## 3 Measurements used by the radiative transfer model to calculate aerosol DRE

### 3.1 Aerosol optical properties

The following aerosol optical properties (including their dependence on wavelength) are standard inputs to RTMs used to calculate aerosol DRE: (1) aerosol optical depth (AOD); (2) single-scattering albedo ($\omega_0$); and (3) scattering asymmetry parameter (g). For calculation of broadband, diurnally averaged aerosol DRE, we form daily averages of each optical property and interpolate or extrapolate to their values at 38 equally spaced wavelengths over the 250 to 4000 nm range. We note that the power-law expressions (Eqs. 1,2, and 4) used to extrapolate aerosol properties measured largely at visible wavelengths to the infra-red may or may not represent their true spectral dependence. However, the solar flux in the infra-red is much less than that in the visible so the simple aerosol spectral parameterizations should be sufficient for broadband DRE calculations.

### 3.1.1 Aerosol optical depth

The CIMEL sunphotometer deployed at APP (known as 'Appalachian_State' within AERONET) measures direct solar radiance at eight wavelengths ($\lambda$=340, 380, 440, 500, 675, 870, 940, and 1020 nm) and sky radiance at four of these wavelengths ($\lambda$=440, 675, 870, and 1020 nm), using standard AERONET protocols (Holben, et al., 1998). The direct solar radiance measurements are used to calculate AOD at each of the eight wavelengths except 940 nm, using the Beer-Lambert-Bouguer equation (Holben, et al., 1998). Direct solar radiance measurements are made at optical air mass intervals of 0.25, corresponding to every ~15 minutes near noon and more often near dawn and dusk. Only Level 2 AERONET AOD

(cloud-screened, calibrated) is used in this study. The uncertainty for Level 2 AOD is small enough (0.01 to 0.015; Eck et al., 1999) so that AERONET serves as 'ground truth' for comparisons with satellite-derived AOD (Levy et al., 2010; Hyer et al., 2011).

Sky radiance measurements made at AERONET sites are used to derive column-averaged aerosol properties including size distributions and $\omega_0$. Single-scattering albedo can only be reliably retrieved to within ~0.03 for AOD ($\lambda$=440 nm) $\geq$ 0.40 (Dubovik et al., 2000). This high loading condition is only satisfied on 2 to 4 days per year at the Appalachian_State site and therefore AERONET $\omega_0$ is not available for use in this study. AOD Ångström exponent ($\text{Å}_{aod}$) in the visible spectral range is typically computed as the slope of a linear fit of log (AOD) versus log ($\lambda$) using available wavelengths between 440 and 870 nm. It is used in this paper to wavelength-scale AOD (Fig.2a) using Eq. (1):

$$AOD(\lambda) = AOD(550nm) \left(\frac{550}{\lambda}\right)^{A_{AOD}^0} \tag{1}$$

### 3.1.2 Single-scattering albedo and scattering asymmetry parameter

The primary aerosol measurements at APP, as part of the NOAA ESRL network, are aerosol light scattering ($\sigma_{sp}$), hemispheric backscattering ($\sigma_{bsp}$), and absorption ($\sigma_{ap}$) coefficients, reported at 450, 550, and 700 nm and for aerosols dried to relative humidity RH$\leq$40% (Sherman et al., 2015). Each of these parameters is measured for both sub-10 µm particles ($PM_{10}$) and sub-1 µm particles ($PM_1$). We use $PM_{10}$ values in this paper, even though $\sigma_{sp}$, $\sigma_{bsp}$, and $\sigma_{ap}$ at APP are dominated by sub-1 µm particles (Sherman et al., 2015). Aerosol optical depth measured in the column is representative of aerosol of all sizes and larger particles can contribute greatly to aerosol light scattering. Thus, $PM_{10}$ optical properties measured near the surface will be most comparable to the column AOD measurements.

A three-wavelength integrating nephelometer (Model 3563, TSI Inc., St. Paul, MN) is used for measurement of $\sigma_{sp}$ (angular range of 7° to 170°) and $\sigma_{bsp}$ (angular range of 90° to 170°). Aerosol light absorption coefficients were determined by a three-wavelength Particle Soot Absorption Photometer (PSAP, Radiance Research, Seattle, WA) up until March 2015. A new light absorption instrument developed at NOAA ESRL (Continuous Light Absorption Photometer, CLAP; Ogren et al., 2013) then

replaced the PSAP, after a one-year inter-comparison of the PSAP and CLAP instruments at APP. The major difference between the CLAP and PSAP is that the CLAP has eight filter spots (versus one for PSAP) and can thus run nearly eight times longer between filter changes that require human intervention. The CLAP-measured $\sigma_{ap}$ values are ~5 to 10 % lower than the PSAP (unpublished result). Aerosols entering the instruments are heated as needed to attain RH $\leq$ 40 % to decouple the influences of aerosol amount and RH on $\sigma_{sp}$, $\sigma_{ap,}$, and $\sigma_{bsp.}$ In-depth discussions of NOAA ESRL aerosol sampling, measurements, and data quality assurance protocols are provided in Sherman et al (2015) and references therein. A scanning humidograph (Sheridan, et al., 2001) is employed at APP to measure the RH dependence of $\sigma_{sp}$ and $\sigma_{bsp}$. The humidograph consists of a humidifier and a second TSI 3563 nephelometer placed downstream of the first nephelometer. A one-hour programmable RH ramp (<40% to 85%) is applied to the air stream entering the second nephelometer. A two-parameter fit of the ratio of humidified aerosol $\sigma_{sp}$ to dried aerosol $\sigma_{sp}$ is applied for each RH ramp to deduce the RH dependence of $\sigma_{sp}$ (Eq.3 of Titos et al., 2016). A similar fit is calculated for $\sigma_{bsp}$. We use this and co-located measurements of RH to scale $\sigma_{sp}$ and $\sigma_{bsp}$ to ambient RH for the dataset used in this paper.

Single-scattering albedo at each of the 38 wavelengths supplied to the Santa Barbara DISORT Radiative Transfer model (SBDART; Ricchiazzi et al., 1998) is calculated by wavelength-scaling the $\sigma_{sp}$ and $\sigma_{ap}$ values at 550 nm, using scattering and absorption Angstrom exponents ($\mathring{A}_{sp}$, and $\mathring{A}_{ap}$), which are calculated from the $\sigma_{sp}$ and $\sigma_{ap}$ values at 450 and 700 nm (Sherman et al., 2015)

$$\omega_0(\lambda) = \frac{\sigma_{sp}(\lambda)}{\sigma_{sp}(\lambda)+\sigma_{ap}(\lambda)} = \frac{\left(\frac{550}{\lambda}\right)^{A_{sp}^0}\sigma_{sp}(550nm)}{\left(\frac{550}{\lambda}\right)^{A_{sp}^0}\sigma_{sp}(550nm)+\left(\frac{550}{\lambda}\right)^{A_{ap}^0}\sigma_{ap}(550nm)} \tag{2a}$$

Dividing the numerator and denominator by $\sigma_{sp}(550\ nm) + \sigma_{ap}(550\ nm)$ allows Eq. (2a) to be re-written solely in terms of the intensive aerosol optical properties $\omega_0$ (550 nm), $\mathring{A}_{sp}$, and $\mathring{A}_{ap}$.

$$\omega_0(\lambda) = \frac{\left(\frac{550}{\lambda}\right)^{A_{sp}^0}\omega_0(550nm)}{\left(\frac{550}{\lambda}\right)^{A_{sp}^0}\omega_0(550nm)+\left(\frac{550}{\lambda}\right)^{A_{ap}^0}\left(1-\omega_0(550nm)\right)} \tag{2b}$$

Spectral $\omega_0$ calculated using Eq. (2b) is displayed graphically for each season in Fig.2b. Radiative transfer models typically only treat the scattering dependence when correcting $\omega_0$ to ambient RH; and assume that absorption changes negligibly with RH. While this approach may or may not hold true for all aerosol types (ex: some organics, sulfur-coated soot), determining the dependence of $\sigma_{ap}$ on RH is experimentally very difficult for all but laboratory studies (especially at high RH) conducted under very controlled conditions (Brem et al., 2012. Thus, we only correct $\sigma_{sp}$ to ambient RH in our corrections of $\omega_0$. Uncertainties in correcting $\sigma_{sp}$ to ambient RH are due to uncertainties in (1) $\sigma_{sp}$ measured by the dry and humidified aerosol nephelometers ($\Delta\sigma_{sp}=9.2\%$, Supplement to Sherman et al., 2015); and (2) RH measured inside the humidified nephelometer ($\Delta RH\sim 3\%$; Titos et al., 2016). Titos et al. (2016) used these values as inputs to a Monte Carlo simulation to estimate the uncertainty in the RH-corrected scattering coefficient as $\Delta\sigma_{sp}\sim 20\%$ (their Fig.2b) for high-RH (>90%) and for moderately hygroscopic aerosols such as those observed at APP (Sherman et al., 2016b). We apply $\Delta\sigma_{sp}\sim 20\%$, along with uncertainty in dried aerosol absorption coefficient ($\Delta\sigma_{ap}=20\%$; Sherman et al., 2015), as inputs to Eq. S9 of supplement to Sherman et al. 2015 to calculate $\Delta\omega_0$. Single-scattering albedo uncertainty is larger for more absorbing aerosols and is zero for purely scattering aerosols ($\omega_0=1$). We use monthly median $\omega_0$ values (Fig.5b) to calculate $\Delta\omega_0\sim 0.03$ for winter and surrounding months and $\Delta\omega_0\sim 0.02$ for summer and surrounding months (Table 2).

Scattering asymmetry parameter is calculated at 450 nm, 550 nm, and 700 nm, based on the hemispheric backscatter fraction $b= \sigma_{bsp} / \sigma_{sp}$ and the parameterization (Andrews et al., 2007)

$$g = 0.9893 - 3.96b + 7.46b^2 - 7.14b^3 \tag{3}$$

Uncertainty in the calculated value of g at ambient RH arises due to uncertainties in the measured $\sigma_{bsp}$ and $\sigma_{sp}$, each of which is subject to the same measurement uncertainties as outlined above. Sherman et al. (2015) reported a nearly identical uncertainty in dried aerosol hemispheric backscatter coefficient ($\Delta\sigma_{bsp}=8.9\%$) as for the scattering coefficient ($\Delta\sigma_{sp}=9.2\%$). This, along with the lack of published uncertainties in humidified $\Delta\sigma_{bsp}$ for similar experimental configurations as that deployed at APP, lead us to use the same uncertainty estimate for ambient-RH $\Delta\sigma_{bsp}$ as for ambient-RH $\Delta\sigma_{sp}$ (~20%). Inserting the ambient-RH uncertainties $\Delta\sigma_{bsp}$ and $\Delta\sigma_{sp}$ into Eq.S8 of supplement to Sherman et al. (2015) lead to

hemispheric backscatter fraction uncertainty $\Delta b \sim 0.0085$, which in turn can be used along with the relation between g and b (Eq.3) to calculate $\Delta g = |\partial g / \partial b| \, \Delta b \sim 0.01$.

Asymmetry parameter is wavelength-scaled to the 38 wavelengths used by SBDART following McComiskey et al. (2008), their Eq. (8):

$$g(\lambda) = g(550nm) \frac{1+\left(\frac{550}{\lambda_g}\right)^2}{1+\left(\frac{\lambda}{\lambda_g}\right)^2} \qquad\qquad\qquad (4)$$

Spectral g calculated using Eq. (4) is displayed graphically for each season in Fig.2c. Following McComiskey et al. (2008), we use 5000 nm for $\lambda_g$. However, McComiskey et al.(2008) noted that the exact value does not significantly alter the calculated DRE values and sensitivities through spectral
dependence of *g,* but ensures physically reasonable results for very small (larger) particle sizes when reaching the Rayleigh (Mie) limit.

An assumption used in this paper is that $\omega_0$ and *g* measured near the surface are representative of these properties in the column, which is typically valid in a well-mixed boundary layer. Most vertical profiles of aerosol normalized relative backscatter measured by the lidar at APP during part of the study period
and afterward (as part of MPLNET) show a qualitatively exponential decay with height and an absence of aerosol layers aloft (unpublished result). In addition, AOD is highly correlated with surface-level aerosol extinction coefficient at APP (r=0.79; Sherman et al., 2016b). These suggest that optical properties may be well represented by measurements made at the surface.

**3.2    Spectral surface reflectance**

The MODIS spectral surface reflectance product (Justice, et al., 2002) is derived from MODIS bands 1 thru 7. These seven bands (B1 thru B7) are centered near 645, 855, 466, 553, 1243, 1628, and 2113 nm, respectively. We use the MODIS Aqua eight-day surface reflectance product (MYD09A1), downloaded
from the Oak Ridge National Laboratory Distributed Active Archive Center (ORNL DAAC), for the DRE studies in this paper. The MYD09A1 product is created by analyzing MODIS spectral observations over eight-day periods and identifying the invariant contributions (i.e., the surface). These products are

gridded, reported at 500 m spatial resolution, and have their own quality assurance and error characteristics. Each MYD09A1 pixel contains the best possible observation (with atmospheric correction applied) during an eight-day period as selected by high observation coverage, low view angle, absence of clouds and cloud shadow, and low aerosol loading (Vermote and Kotchenova, 2008). For each eight-day

MYD09A1 product, we calculate the mean surface reflectance of all 500 m pixels in a 10 km x 10 km box (corresponding to 20 x 20 pixels) centered at the APP site, for each of the seven MODIS bands. Only eight-day surface reflectance products with at least 50% of pixels in the 10 km x 10 km box passing MODIS quality assurance tests are used in this study. Because surface reflectance varies primarily on seasonal timescales, we form monthly averages at each wavelength from the eight-day products.

Uncertainty in the MODIS surface reflectance product under low aerosol loading conditions is 5 % or $5.0*10^{-4}$, whichever is larger (Vermote and Saleous, 2006). Surface reflectance at APP is always large enough so that $\Delta R=0.05*R$. Since R is wavelength dependent for any surface type, the uncertainty $\Delta R$ depends on wavelength. To set an upper bound on $\Delta R$, we note that monthly averaged R at APP is highest in summer (Fig.3b) and for the 855 nm band (B1) and the 1243 nm band (B5), with summer values near

0.40. We use this to estimate $\Delta R \sim 0.02$ for this study. For simplicity, we neglect the wavelength dependence of $\Delta R$, which if considered would result in a smaller $\Delta R$ (Figs.3 a-d).

The SBDART radiative transfer model used in this study to calculate DRE parameterizes spectral surface reflectance as a linear combination of that due to vegetation, sand, water, and snow, based on user-provided coefficients specifying the contributions due to each surface type. For each month, we

calculate spectral surface reflectance in SBDART for a range of relative vegetation, sand, water, and snow contributions and select the combination which minimizes the mean-square difference (weighted by relative solar irradiance) with the monthly averaged MODIS spectral surface reflectance. Note that the relative contributions from the different surface types do not need to add up to 1.0. As an example, the mean spectral surface reflectance measured by MODIS above the APP site in June most closely

approximates that of 0.75 times the spectral reflectance curve produced by vegetation alone, with no contributions from the sand, water, or snow spectral reflectance curves (Fig.3b). This is due to the darker vegetation from heavy deciduous forest in the region surrounding the site.

## 4 Methodology

### 4.1 SBDART radiative transfer model used to calculate DRE

To calculate clear sky, broadband (250 to 4000 nm) aerosol DRE, we run the SBDART model at 5 nm spectral resolution, with clouds and stratospheric aerosols turned off. We configure the model to use four radiation streams (i.e. four zenith and four azimuthal angles), which provides a good combination of computational efficiency and accuracy for calculating fluxes (Ricchiazzi et al. 1998). We use the LowTran-7 atmospheric transmission model, which possesses 20 cm$^{-1}$ resolution. Standard midlatitude summer vertical profiles of pressure, temperature, water vapor density, and ozone density (McClatchey et al., 1972) are used for April thru October and standard mid-latitude winter profiles are used for November thru March. Although some differences from the actual vertical meteorological profiles are expected, the same standard vertical meteorological profiles are used to calculate the radiative fluxes with and without aerosols and thus would not be expected to contribute much to the calculation of DRE, which is based on the difference in fluxes (Eq. (5)). The aerosol scattering asymmetry parameter (Sect. 3.1.2) supplied to SBDART is used to estimate aerosol scattering phase function, in the Henyey-Greenstein approximation (Henyey and Greenstein, 1941). Vertical distribution of aerosols is believed to be a second-order effect in the calculation of aerosol DRE for primarily scattering aerosols (McComiskey et al., 2008) and we use the SBDART default vertical aerosol density profile in this initial study. The default profile uses an assumed exponential decrease in aerosol density with a scale height inversely proportional to surface-level aerosol light extinction coefficient at 550 nm (Ricchiazzi et al., 1998), which is calculated as the sum of the measured $\sigma_{sp}$ and $\sigma_{ap}$ (Sect. 3.1.2). The overall curve is scaled by the AOD (Sect.3.1.1). Aerosol density scale heights used by SBDART range from 1.05 to 1.51 km, which qualitatively agree with typical MPL-measured normalized relative backscatter profiles under clear sky conditions at APP (Sect. 2).

### 4.2 Seasonal variability in aerosol optical properties and DRE

For the study of seasonal DRE variability (Sect. 5.1), we use the SBDART model to calculate diurnally averaged DRE at the TOA and at the surface, for 418 days during the period 14 June 2012 thru 28 June 2016. We then bin the DRE by month and calculate statistics for each month (Figs. 4a and 4b). For each

of the 418 days, we calculate DRE for each hour to account for the effect of varying solar geometry on the calculation of diurnally-averaged DRE. For each hour, we supply daily-averaged AOD($\lambda$), $\omega_0(\lambda)$, and g($\lambda$) as inputs to the SBDART model. We also supply the coefficients specifying the best-fit linear combination of surface types (snow, water, sand, vegetation) to the MODIS monthly-averaged spectral

surface reflectance (Figs.3; Sect. 3.2). Upwelling and downwelling broadband shortwave fluxes for that hour are calculated with average measured aerosol properties and then with no aerosols and their difference is used to calculate DRE using Eq. (5)

$$DRE = (F_{A\downarrow} - F_{A\uparrow}) - (F_{NA\downarrow} - F_{NA\uparrow}) \tag{5}$$

The process is repeated for all 24 hours and the results averaged to yield diurnally averaged DRE. Since

AOD is only measured during daytime hours, the daily-averaged AOD used as RTM input may or may not be representative of AOD during night-time hours.  However, AOD during night-time hours does not affect the calculation of shortwave solar fluxes, since these fluxes (both with and without aerosols) are zero during night-time (leading to calculated DRE=0 for these hours). In addition to DRE, we calculate the aerosol direct radiative efficiency (RE) by dividing diurnally averaged DRE by daily averaged AOD

at 550 nm.  Radiative efficiency is to first-order independent of aerosol amount (i.e. AOD), and dependent on the inherent optical nature of the aerosol, controlled by composition and size. It is a useful quantity for determining whether DRE varies due to changes in aerosol loading or aerosol type. Use of daily averaged DRE in this study integrates over solar angles and the use of daily averaged aerosol optical properties is justified by the small diurnal variability in AOD, $\omega_0$, and g at APP (Figs.9), although diurnal aerosol

variability does introduce uncertainty into the DRE calculations (Sect. 5.4).).

## 4.3    Sensitivity of aerosol DRE to aerosol properties and surface reflectance

To study the sensitivity of clear sky aerosol DRE to changes in AOD, $\omega_0$, g, and R (Sect. 5.2), we follow a similar approach to that used by McComiskey et al (2008). We define the sensitivity $S_i$ of diurnally averaged DRE to parameter $\rho_i$ (where $\rho_i$ stands for either AOD, $\omega_0$, g, or R) as the change in

DRE per unit change in $\rho_i$. Formally, $S_i$ is evaluated as the partial derivative of DRE with respect to $\rho_i$ ($S_i = \partial (DRE) / \partial \rho_i$) evaluated at the 'base case' values for all variables (Table 1). To assess whether these sensitivities $S_i$ are independent of the respective $\rho_i$ values, we plot diurnally averaged DRE versus $\rho_i$ over

the largest expected range of each $\rho_i$ at APP (Table 2; Figs. 6 and 7). For the sensitivity $S_R$, we scale the entire spectral surface reflectance curve (Figs.3) by proportionally scaling the input surface type coefficients supplied to SBDART (Figs.3), to vary thebroadband (250-4000nm) surface reflectance R (Figs. 6d and 7d). For example, doubling both the sand and vegetation coefficient values supplied to

SBDART scales the entire September surface reflectance curve (Fig.3c) by the same amount, thereby doubling the base-case value of broadband R in Table 1. Insensitivity of DRE to AOD, $\omega_0$, g, or R is inferred from the degree of linearity of the respective plot (i.e. a constant slope $S_i$).

Since aerosol optical properties (Figs.5) and surface reflectance (Figs.3) at APP vary primarily on seasonal scales, we evaluate the $S_i$ separately for each season. It is impractical (and unnecessary) to

construct sensitivity curves for each individual month so we choose one representative calendar day to represent each season; December 21 for winter, March 21 for spring, June 21 for summer, and September 21 for fall. We refer to these seasonally-representative days as DEC, MAR, JUN, and SEP. Inclusion of the equinox days (with equal durations of sunlight and darkness) also provides results which may possibly be indicative of 'annually averaged values'. We use monthly median AOD, $\omega_0$, and g values at 550 nm

as base case values but monthly mean values are similar to medians at APP and could also be used with negligible difference in results (Figs.5 a-c). Spectral dependence of each aerosol property is calculated from the values at 550 nm, using the approach of Sect(s). 3.1. The base case R values in Table 1 are the broadband surface reflectance corresponding to the monthly mean spectral surface reflectance curves (Figs.3). We then vary the independent variables $\rho_i$ individually about these base case values (Table 1)

to generate the 'seasonal' DRE versus $\rho_i$ curves. We evaluate $S_i = \partial\,(DRE)\,/\,\partial\rho_i$, at base case $\rho_i$ value, as the regression slope of the five points on each side of the base case value.

## 4.4    Estimating uncertainty in aerosol DRE

The DRE sensitivity values $S_i = \partial RE/\partial p_i$ and known (or estimated) measurement uncertainties, $\Delta AOD$, $\Delta\omega_0$, $\Delta g$, and $\Delta R$ (Table 2) can be used to calculate the corresponding uncertainty in DRE using Eq. (6)

$$\Delta DRE = \sum_{i=1}^{4} \sum_{j=1}^{4} \frac{\partial DRE}{\partial \rho_i} \frac{\partial DRE}{\partial \rho_j} \text{cov}(\rho_i, \rho_j) = \sum_{i=1}^{4} \sum_{j=1}^{4} \frac{\partial DRE}{\partial \rho_i} \frac{\partial DRE}{\partial \rho_j} \text{corr}(\rho_i, \rho_j)\, \Delta\rho_i \Delta\rho_j$$

(6)

Where $cov(\rho_i, \rho_j)$ is the covariance of $\rho_i$, and $\rho_j$, which in turn can be expressed in terms of their linear correlation $corr(\rho_i, \rho_j)$. The double summation 'i' and 'j' is over the four RTM input parameters (AOD, $\omega_0$, g, and R). The summations in Eq. (6) can be explicitly written as

$$\Delta DRE^2 = S_{AOD}^2 \Delta AOD^2 + S_{\omega_0}^2 \Delta \omega_0^2 + S_g^2 \Delta g^2 + S_R^2 \Delta R^2 + 2 S_{AOD} S_{\omega_0} corr(AOD, \omega_0) \Delta AOD \Delta \omega_0 +$$

$$2 S_{AOD} S_g corr(AOD, g) \Delta AOD \Delta g + 2 S_{AOD} S_R corr(AOD, R) \Delta AOD \Delta R + 2 S_{\omega_0} S_g corr(\omega_0, g) \Delta \omega_0 \Delta g +$$

$$2 S_{\omega_0} S_R corr(\omega_0, R) \Delta \omega_0 \Delta R + 2 S_g S_R corr(g, R) \Delta g \Delta R \tag{7}$$

The first four terms of the sum in Eq. (7) facilitate estimates of the contributions to ΔDRE due to the individual sources of uncertainty, neglecting covariance effects (McComiskey et al., 2008). However, correlations amongst some aerosol optical properties are non-negligible at APP during some seasons (Table 3) and must be considered for an improved estimate of ΔDRE. We calculate ΔDRE both with and without the inclusion of the covariance terms to examine their effect on ΔDRE. An equation identical to Eq. (7) can be used to calculate the uncertainty in RE. Results from a similar analysis of the sensitivity of RE to AOD, $\omega_0$, g, and R, along with the associated uncertainties in RE, are provided in Sect(s). S1 and S2 of the supplement to this paper.

## 5    Results and discussion

### 5.1    Annual cycles of aerosol DRE, RE, and aerosol optical properties

Aerosol DRE and the optical properties influencing DRE demonstrate large seasonal variability at APP. Median diurnally averaged DRE for the nearly four-year study period is -2.9 Wm$^{-2}$ at the TOA and -6.1 Wm$^{-2}$ at the surface (Figs. 4a and 4b). Median DRE at the TOA is ~5 to 6 times larger (i.e. more negative) during summer months (JJA) than during winter months (DJF) and median DRE at the surface is ~4 times larger in summer than in winter. Median DRE at both the TOA and surface is nearly twice as negative during summer months as the median for the entire period. Variability in DRE is also largest during summer and the surrounding months (Figs. 4a and 4b). While fewer daylight hours during winter obviously contributes to the seasonal differences in diurnally averaged DRE, the annual DRE cycles clearly follow that of AOD (Fig.5a). Median and mean AOD at 550 nm are approximately 9 to 10 times

larger in summer than in winter and AOD variability is also largest during summer and the surrounding months.

Further confirmation of the dominant influence of AOD on the annual DRE cycles at APP is seen in the annual RE cycles. Aerosol RE is most sensitive to $\omega_0$, followed by comparable sensitivities to g and R. It is least sensitive to AOD (Table.S1 of Supplement to this paper). Based on this, the annual DRE and RE cycles should be qualitatively similar if $\omega_0$ and/or g (rather than AOD) exert the primary aerosol influences on the DRE cycle. Instead, median and mean RE are more negative during winter months than during summer months by a factor of ~ 1.5 at the TOA and ~1.5 to 2 at the surface (Figs. 4c and 4d) and RE variability is also largest during winter months. Thus, months with the most negative DRE coincide with the least negative RE. One exception is April, for which median and mean AOD are only half of that during the summer months but surface DRE (and to a lesser extent TOA DRE) is close to that during summer months. April is characterized by a relatively low $\omega_0$ (Fig.5b) and low g (Fig.5c), in addition to a darker surface during spring than summer (Figs.3a and 3b). This leads to a similar RE to that of winter months, coinciding with a high enough AOD to produce surface DRE close to that during summer.

The annual RE cycles at both the TOA and the surface (Figs. 4c and 4d) can be explained using the following information: (1) the signs of the sensitivities of RE to increases in g and R are positive at both the TOA and surface while the sensitivity of RE to increases in $\omega_0$ is positive at the surface and negative at TOA (Figs. S1 and S2 of supplement to this paper ); and (2) $\omega_0$, g, and R are all larger during the warm season (summer and surrounding months), with smaller values during most other months (Figs. 3a-d, 5b, and 5c). The fact that the RE sensitivities to $\omega_0$, g, and R are all positive at the surface (i.e. an increase in each of these parameters drives RE at the surface to less negative values) results in less negative surface RE values during summer and more negative RE values during winter and the surrounding months. The annual RE cycle at the TOA appears to be influenced more by the combined influences of g and R than by $\omega_0$, since the decreases in g and R as one moves away from the summer months drives the TOA RE more negative than the positive influence on RE due to lower $\omega_0$ (Fig.4c). This is despite greater sensitivity of RE to $\omega_0$ and is likely due to the larger (by a factor of ~2) summer-winter differences in g and R than in $\omega_0$ (Figs. 5b and 5c).

Alston and Sokolik (2016) reported a mean TOA DRE of approximately -10 Wm$^{-2}$ for mountainous western North Carolina and surrounding areas during summer (their Fig.7), with somewhat more negative values (-12 to -16 Wm$^{-2}$) for much of the SE U.S. Their DRE values appear to be calculated based on cloud fraction of ~40 % (their Fig.2) while ours are for clear sky conditions (i.e. cloud fraction of zero)

so we need to multiply their DRE values by a factor of 1 / (1-0.40) ~1.7 to compare with our clear sky DRE. Our monthly mean clear sky TOA DRE of -5 to -6 Wm$^{-2}$ during summer months (Fig.4a) is approximately three times smaller than Alston and Sokolik's clear sky values (-17 Wm$^{-2}$). The large DRE difference cannot be explained solely by differences in the aerosol optical properties and R used in the calculations. Alston and Sokolik appear to have used the following values in their TOA DRE calculation:

(a) AOD ($\lambda$=550 nm) ~0.25-0.30 (their Fig.1); (b) $\omega_0$ ($\lambda$=558 nm) = 0.96; and (c) R~0.145 (their Fig.3). They did not state the value of g used (related to the upscatter fraction in the Haywood and Shine, 1995 equation 2). Using our TOA DRE sensitivity parameters (Table 3) and summer base case values (Table 1), the difference in AOD between the studies (0.25 versus 0.15) gives rise to a difference in TOA DRE of only ~4 Wm$^{-2}$. The differences in R (0.21 versus 0.145) and $\omega_0$ values (0.96 versus 0.95) lead to an

additional TOA DRE difference of ~1.5 Wm$^{-2}$. When added together, these account for approximately half of the TOA DRE discrepancy.

A likely source for the other half of the large summer TOA DRE discrepancy is a difference in methods used to calculate TOA DRE. Inserting summer base case values at APP (Table 1) into the first-order DRE equation used by Alston and Sokolik (Eq. (2) of Haywood and Shine, 1995) leads to summer TOA DRE

of ~ -10 Wm$^{-2}$ at APP, which is approximately twice as negative as monthly mean (and median) TOA DRE calculated using the SBDART RTM (Fig.4a). The Haywood and Shine (1995) equation used by Alston and Sokolik (2016) is valid for an optically thin atmosphere and uses spectrally weighted aerosol optical properties and surface reflectance as inputs. The degree to which the first assumption holds obviously decreases with the higher AOD of summer months while the degree to which aerosol optical

properties at 550 nm are representative of spectrally weighted properties also impacts the resultant TOA DRE. Using an identical procedure to that outlined above for summer, our winter DRE values using the simplified Haywood and Shine (1995) TOA DRE equation are approximately the same as those calculated using the SBDART model. Unlike summer, a majority of the (~2 to 3 Wm$^{-2}$) difference between our

monthly mean TOA DRF and that reported by Alston and Sokolik (2016) is consistent with differences in AOD and cloud fraction between the studies. Sensitivity of TOA DRF to $\omega_0$, g, and R is so small during winter (Table 3 and Sect. 5.2) that differences in these properties is unlikely to influence TOA DRF agreement.

## 5.2    Sensitivity of DRE to aerosol optical properties and surface reflectance

The plots of aerosol DRE versus AOD, $\omega_0$, g, and R (Figs. 6 and 7) are for the most part linear, indicating that the sensitivities (i.e. slopes of plots) are independent of the values of aerosol optical properties and surface reflectance, at least over the ranges observed at the APP site. There are a few minor exceptions, namely (1) sensitivity of TOA and surface DRE to AOD for high AOD values in DEC and MAR; and (2) TOA and surface DRE sensitivity to $\omega_0$ and R in JUN and SEP. The nonlinearity in DRE versus AOD leads to a small dependence of RE on AOD, more so in DEC and MAR (See also Table S1 of Supplement). However, AOD values during the non-summer months at APP are rarely large enough (Fig.5a) to lie on the nonlinear portion of the curves (Figs. 6a and 7a).

Aerosol DRE at APP is most sensitive to changes in AOD, followed by $\omega_0$ (Table 3). The sensitivities $S_{\omega o}$ and $S_{AOD}$ are comparable during summer (JUN) and fall (SEP) at the TOA but $S_{AOD}$ is much larger than $S_{\omega o}$ during the months with lower AOD ($\leq 0.05$; DEC and MAR). Aerosol DRE is less sensitive to changes in g and R. The sensitivity $S_{AOD}$ is greatest for MAR (-47 Wm$^{-2}$ AOD$^{-1}$ at TOA and -90 Wm$^{-2}$ AOD$^{-1}$ at surface) but $S_{AOD}$ at the TOA and surface exhibit only modest variation with season (~25 to 30 %).  Values of $S_{AOD}$ at the surface are close to twice those at the TOA for all seasons. The sensitivities $S_{\omega o,}$, $S_g$, and $S_R$ vary much more with season than does $S_{AOD}$ (Table 3), with values 8 to 15 times greater in JUN than in DEC. Higher sensitivity of DRE to $\omega_0$, g, and R  for higher AOD conditions was also reported by McComiskey et al. (2008). The fact that $S_{\omega o}$ is negative at the TOA and positive at the surface (Table 3; Figs.6b and 7b) implies that increasing $\omega_0$ leads to a larger cooling effect at the TOA and a smaller cooling effect at the surface. The magnitude of $S_{\omega o}$ is ~40 to 60 % greater at the surface than the TOA for all seasons. In contrast, the magnitudes of $S_g$ are nearly identical at TOA and surface for all seasons. The fact that the signs of $S_g$ are positive at both the TOA and surface is consistent with larger particles (corresponding to larger g) scattering a greater fraction of light in the forward direction than do

smaller particles (smaller g). Relatively low sensitivity of DRE to surface reflectance at APP during the study period is seen by the low $S_R$ values, which range from 17 Wm$^{-2}$ (14 Wm$^{-2}$) at the TOA (surface) during JUN to 2.0 (1.8) Wm$^{-2}$ at TOA (surface) during DEC, per unit change in R (Table 3).

5     The only similar study of DRE sensitivities and uncertainties to include a continental U.S. site was that of McComiskey et al. (2008), which included the Southern Great Plains DOE ARM site (SGP; Lamont, OK) and the month of SEP. Alston and Sokolik (2016) investigated the effects of changes in AOD, R, and cloud fraction on TOA DRE and concluded that DRE changed most in response to changes in AOD. However, they did not derive numerical values for the sensitivities. McComiskey et al. (2008) used similar aerosol optical properties at 550 nm to our SEP values (Table 1) in estimating DRE sensitivities at the 10 SGP site (AOD=0.10; $\omega_0$=0.95; g=0.60), leading to sensitivities that were all within ~20% of our SEP sensitivities, at both the TOA and the surface (Table 3). While TOA $S_R$ values at APP (our study) and at SGP (McComiskey et al., 2008) are only ~10-20% larger than $S_R$ values at the surface, Gadhavi and Jayaraman (2004) reported much higher DRE sensitivity to surface type at the TOA than at the surface in Antarctica. As part of their analysis, Gadhavi and Jayaraman (2004) plotted both TOA and surface DRE 15 versus AOD for different surface types (snow, seawater, sand, and vegetation), using the same SBDART radiative transfer code used in our study and by McCommiskey et al. (2008). Their TOA DRE changed by ~10 W m$^{-2}$ as they changed the surface type from all sea water to all snow (their Fig.10, for AOD=0.10), which represents close to a unit change in surface albedo. However, their corresponding change in surface DRE was only ~3 W m$^{-2}$.

20     To gain more insight into the importance of AOD, $\omega_o$, and the underlying surface type on DRE at the TOA and surface (including their seasonally-dependent sensitivities), we plot DRE (at surface and TOA) versus AOD and DRE versus $\omega_0$ for each surface type (snow, seawater, sand, and vegetation) in Figs.8. We only include plots for JUN and DEC (high and low aerosol loading, respectively). Plots for all seasons are included in the supplement to this paper (Figs. S3 and S4). For a purely-scattering aerosol ($\omega_o$=1), 25 DRE at the TOA and surface are always equal, regardless of surface type and AOD (Figs.8 a and b). This is due to the absence of atmospheric aerosol light absorption. An increase in aerosol light absorption resulting from a darker aerosol (lower $\omega_o$) and/or increase in AOD (for $\omega_o$<1) always gives rise to a more negative DRE at the surface but the directional change in TOA DRE depends on the relative albedos of

the aerosol and the underlying surface (Figs. 8a-d). Increases in aerosol light absorption always leads to a larger difference between surface and TOA DRE, for all surface types (Figs.8a-d). For a fixed AOD, the difference between TOA and surface DRE is largest for a darker aerosol above a brighter surface and is smallest for a brighter aerosol above a darker surface (Figs.8 a and b), due to increased aerosol

absorption of light reflected from a brighter surface and making a second pass through a more absorbing atmosphere (Chung, 2012). Increased aerosol light absorption is also the reason for higher sensitivity of DRE to small changes in $\omega_o$ (i.e. larger $S_{\omega o}$) for higher AOD conditions, such as summer months at APP (AOD≥0.15). The aerosols at APP (and at SGP; McComisky et al., 2008) are primarily scattering ($\omega_0$~0.91-0.95) and the surface is relatively dark (R~0.12-0.21; Table 1) so $S_R$ and its differences between

TOA and surface would not be expected to be large. As with $S_{\omega o}$, higher AOD leads to higher $S_R$ values during summer months than during winter months at APP (Table 3; Figs. 6d and 7d). We speculate (based on Figs. 8 a and b) that the much higher DRE sensitivity to surface type at the TOA than at the surface reported by Gadhavi and Jayaraman (2004) may be due to more absorbing aerosols (lower $\omega_0$) in their study. Their DRE sensitivity curves (with respect to AOD) used the same AOD range as ours (Figs. 8 c

and d) but they did not report the value of $\omega_0$ (or its spectral dependence) used to generate their sensitivity curves (their Fig.10).

## 5.3   DRE measurement uncertainties

Uncertainty in aerosol DRE at both the TOA and surface is largest in JUN and lowest in DEC (Tables 4 and 5), primarily due to the highest base case AOD in summer (JUN) and lowest in winter (DEC) (Table

1). Fractional DRE uncertainties are highest in winter (DEC), when AOD and DRE are smallest (Figs.4 a-b and 5a). Uncertainties in diurnally averaged DRE at the TOA (surface) are 20 to 24% (15 to 22 %) of the DRE calculated using base case values for the given month (Table 1), except for DEC, when the DRE uncertainty reaches 50 % at both the TOA and surface. Since the measurement uncertainties of AOD, $\omega_0$, g, and R are all between 0.01 and 0.03 (Table 2), the relative contributions of each parameter to the total

DRE uncertainty are largely influenced by their DRE sensitivity values (Table 3).

Uncertainty in DRE at both the TOA and the surface is dominated by AOD uncertainty for the months with lowest base case AOD (DEC and MAR). In contrast, DRE uncertainty is most influenced by

uncertainty in $\omega_0$ during higher loading summer months (JUN; AOD$\geq$ 0.15), due to comparable values of $S_{\omega o}$ and $S_{AOD}$ (Table 3) coupled with higher measurement uncertainties in $\omega_0$ (Table 2). Both AOD and $\omega_0$ contribute approximately equally to DRE uncertainties during the intermediate-loading month of SEP (base-case AOD=0.10). Uncertainties in g and R contribute less than AOD and $\omega_0$ uncertainties to the

DRE measurement uncertainty during all seasons. Inclusion of covariance effects in the DRE uncertainty calculations increases the JUN and SEP DRE uncertainty at the TOA (surface) by approximately 0.1 to 0.2 W m$^{-2}$ (0.3 W m$^{-2}$), due to the higher correlations amongst AOD, $\omega_0$, and g during the warm-season months at APP (Table 3). It is interesting to note that $\omega_0$ and g at are highly correlated during all seasons, with corr($\omega_0$,g) between 0.78 and 0.85 (Table 3). This indicates a strong tendency for larger particles at

APP to be more reflective. However, the sensitivities $S_{\omega o}$ and $S_g$ are only large enough to lead to non-negligible covariance effects during the higher aerosol loading months (JUN and SEP) at APP (Eq. (7), Tables 4 and 5).

Due to the wide usage of MODIS-measured AOD in aerosol DRE studies, it is instructive to compare DRE uncertainties calculated using AERONET AOD with those using MODIS AOD. In their global

inter-comparison of MODIS Collection 5 AOD with AERONET, Levy et al. (2010) estimated the MODIS AOD error envelope to be $\pm(0.05+0.15*AOD_{aeronet})$ over land. For our comparison of DRE uncertainties, we use the lower MODIS AOD uncertainty $\Delta AOD_{MODIS}$=0.05 in Eq. (7), in place of the AERONET AOD uncertainty of 0.01 (Table 2). Sherman et al (2016a) reported excellent correlation of MODIS AOD and daily averaged AERONET AOD above APP so we use the same correlation values to calculate the

covariance terms involving AOD in Eq. (7) (Table 3). Uncertainty in TOA DRE calculated using MODIS AOD is between 2.0 and 2.3 Wm$^{-2}$ for each season (Table 4), which corresponds to 39 % of the base case JUN DRE and 240 % of the DEC base case DRE (Table 4). In terms of absolute DRE uncertainties, those based on MODIS AOD range from 2 to 5 times the DRE uncertainties using AERONET AOD. Similar fractional DRE uncertainties as that at the TOA are obtained at the surface (Table 5), which correspond

to DRE uncertainties ranging from 3.5 Wm$^{-2}$ (SEP) to 4.5 Wm$^{-2}$ (MAR). Due to the higher uncertainty in MODIS AOD (relative to AERONET), the AOD uncertainty is the dominant term in DRE uncertainty for all seasons when MODIS AOD is used. It should be noted that an AOD error envelope for the new

MODIS collection (C6) is not yet available but it could be smaller than that of Collection 5, given algorithm improvements (Levy et al., 2013).

## 5.4 DRE uncertainty due to diurnal variability in aerosol optical properties

The use of daily-averaged aerosol optical properties as inputs to the RTM can contribute to DRE uncertainty at sites with large diurnal aerosol variability. We apply the DRE sensitivity parameters (Sect. 5.2) to estimate uncertainties in diurnally-averaged DRE resulting from diurnal variability in AOD, $\omega_0$, and g. To estimate diurnal aerosol variability at APP, we apply a similar technique to that used by Sherman et al. (2015) and Sherman et al. (2016a). We form hourly averages of AOD, $\omega_0$, and g and then bin them by hour of the day, for each season. We calculate the mean for each hour of the day and use standard error of the mean to assess whether the diurnal variability in mean values are statistically-significant (Figs.9). We estimate the uncertainty in AOD, $\omega_0$, or g as the amplitude of the diurnal cycle in mean values (Figs.9), relative to the daily-mean value. Calculations identical to those used to estimate DRE measurement uncertainty (Sect. 5.3) are then performed to estimate DRE uncertainty due to the use of daily-averaged aerosol properties (Table 6). We note that the uncertainties would be larger for satellite-based DRE estimates, if the values of aerosol properties retrieved at time of overpass differ from daily-mean values.

Diurnal variability in $\omega_0$ is less than $\omega_0$ measurement uncertainty for all seasons (Fig. 9b; Table 6). In contrast, diurnal variability in g (Fig.9c; Table 6) exceeds measurement uncertainty for all seasons except winter, when they are of equal magnitudes ($\Delta g \sim 0.01$). The diurnal cycle in ambient-RH g follows that of RH and is not observed in the dried aerosol optical properties at APP (Sherman et al., 2015). Diurnal variability in AOD is less than or equal to AOD measurement uncertainty in winter and spring, with larger values in summer and fall. Aerosol diurnal variability leads to similar DRE uncertainties (Table 6) as that due to measurement uncertainties during DEC and MAR, due to the primary sensitivity of DRE to AOD during low-loading months, along with similar values of $\Delta$AOD due to measurement uncertainty and diurnal variability. Diurnal variability contributes to diurnally-averaged DRE uncertainties that are ~20-30% greater than those calculated using measurement uncertainties during JUN and SEP, for both the TOA and surface (Table 6). Diurnal variability in AOD and g exceeds the corresponding measurement uncertainties during these months and the resulting impact on DRE uncertainty more than offsets the

greater sensitivity $S\omega_0$. Note that this would not hold true if satellite-retrieved AOD is used in place of AERONET AOD. In this case, DRE uncertainty due to measurement uncertainties would exceed that due to aerosol diurnal variability for all seasons at APP.

## 6    Summary and conclusions

Daily-averaged aerosol optical properties measured at Appalachian State University's co-located NASA AERONET and NOAA ESRL aerosol monitoring sites over a nearly four-year period are used along with monthly averaged spectral surface reflectance measured by MODIS to study the annual cycles of diurnally averaged clear sky aerosol DRE and RE. This study is the first multi-year 'ground truth' DRE study in the SE U.S., using aerosol network data products that are often used to validate satellite-based

aerosol retrievals (Levy et al., 2010; Sherman et al., 2016a). The study is also the first in the SE U.S. to quantify DRE uncertainties and sensitivities to aerosol optical properties and surface reflectance, including their seasonal dependence.

Median diurnally averaged clear sky DRE at APP over the study period is -2.9 $Wm^{-2}$ at the TOA and -6.1 $Wm^{-2}$ at the surface. Monthly median and mean DRE at the TOA (surface) range from -1 to -2 $Wm^{-2}$

($-2$ to -3 $Wm^{-2}$) during winter months to -5 to -6 $Wm^{-2}$ (negative 10 $Wm^{-2}$) during summer months. While the annual DRE cycle at APP largely follows that of AOD, aerosol RE demonstrates an anti-correlation with AOD and DRE. The most negative RE is observed during November thru April at the TOA and during October thru April at the surface. The least negative RE is observed in June thru September at both the TOA and the surface. Aerosol DRE is most sensitive to changes AOD, followed

by $\omega_o$. It is less sensitive to g and R. One exception is that the sensitivity of TOA DRE with respect to $\omega_o$ is comparable to that of AOD during summer (JUN), when the base case AOD at APP is highest. Since the measurement uncertainties in AOD, $\omega_o$, g, and R are of comparable magnitude, the relative contributions of each to the total DRE uncertainty are largely influenced by their DRE sensitivities. The sensitivities $S_{\omega o}$, $S_g$, and $S_R$ vary by factors of ~8 to 15 with season and are largely influenced by AOD.

In contrast, $S_{AOD}$ exhibits much more modest seasonal variation (~25 to 30 %). This result supports the assertion that the seasonal dependence of $S_{\omega o}$, $S_g$ (and to lesser extent, $S_R$) must be accounted for in DRE uncertainty estimates, at least for sites like APP where there is large seasonality in aerosol loading and in dominant aerosol types.

Using seasonally representative aerosol optical properties and surface reflectance from APP, clear sky aerosol DRE uncertainty at the TOA (surface) due to measurement uncertainties in the RTM inputs ranges from 0.45 Wm$^{-2}$ (0.75 Wm$^{-2}$) for DEC to 1.1 Wm$^{-2}$ (1.6 Wm$^{-2}$) for JUN. Expressed as a fraction of DRE calculated using base case aerosol optical properties and surface reflectance (Table 1), the DRE uncertainties at the TOA (surface) are 20 to 24 % (15 to 22 %) for MAR, JUN, and SEP and 50%  for DEC. Unlike the McComiskey et al.(2008) study, we include the effect of covariances amongst aerosol optical properties in order to determine their effect on DRE uncertainty. Covarience impacts on DRE uncertainty at APP are negligible for low AOD conditions (AOD$\leq$0.05 at 550nm) during winter and surrounding months but do increase $\Delta$DRE by ~0.1 to 0.3 Wm$^{-2}$ under moderate and high AOD conditions (AOD$\geq$0.10 at 550nm) during summer and surrounding months. Uncertainty in diurnally-averaged DRE also arises due to the use of daily-averaged aerosol optical properties as inputs to the RTM. Though diurnal aerosol variability in AOD, g, and especially $\omega_0$ is relatively small above APP (<0.01 to 0.03), this variability can lead to DRE uncertainties comparable in magnitude to those resulting from measurement uncertainties during winter and spring and ~20-30% larger in summer and fall. The primary reason for the relatively low DRE uncertainties reported in this study is the small uncertainty in AOD measured by Cimel sunphotometers as part of AERONET (Eck et al., 1999). The DRE uncertainties are dominated by AOD uncertainties for all seasons when MODIS AOD is used as the AOD input to the radiative transfer calculations, with DRE uncertainties 2 to 5 times larger than those using AERONET AOD.

The results from our study suggest that while satellite-based aerosol measurements provide the necessary global coverage for climate studies, their current levels of uncertainty necessitate complementary ground truth measurements of AOD, $\omega_0$, and g (or some other proxy for scattering phase function) from regionally representative sites to better constrain aerosol DRE in models. Continuous, long-term aerosol measurements from ground-based aerosol network sites are necessary not only to evaluate satellite-based aerosol retrieval accuracy but also to assess whether AOD at the time of satellite overpass is representative of daily averaged AOD at the site (Sherman et al., 2016a). Our study also illustrates the challenges faced in such 'ground truth' DRE estimates. Aerosol optical depth over much of the non-urban U.S. is too low to retrieve column-averaged $\omega_0$ with an uncertainty less than ~0.03 (at best)

at AERONET sites, under most conditions (Dubovik, et al., 2000). Ground-based in situ aerosol networks such as NOAA ESRL can measure dried aerosol $\omega_0$ with an uncertainty ~0.015 (Sherman et al., 2015) even under these low-loading conditions but the dried aerosol properties must be corrected to ambient RH, which requires knowledge of the hygroscopic dependence of aerosol light scattering. Currently such

measurements exist at only two NOAA ESRL sites in the U.S. (APP and SGP). In addition, some knowledge of vertical aerosol profiles is necessary to assess whether the near-surface aerosol properties are likely representative of the column-averaged aerosols. Though qualitative inspection of lidar-measured aerosol backscatter profiles at APP indicates that aerosols are generally confined to the first 1 to 2 km, the availability of multi-year, quality-assured profiles of aerosol light extinction (as part of

MPLNET) will facilitate expansion of this initial DRE study to include the effects of vertical aerosol structure on DRE at APP.

**Data availability:** AERONET Level 2 aerosol optical depth data used in this paper is publicly available at the NASA AERONET website (https://aeronet.gsfc.nasa.gov/cgi-bin/webtool_opera_v2_new). The

aerosol measurements made as part of NOAA ESRL are publicly available for download at the Global Atmospheric Watch World Data Centre for Aerosols (http://ebas.nilu.no/Default.aspx) and graphical representations of the data are available at the NOAA ESRL website (https://www.esrl.noaa.gov/gmd/aero/net/status_plot.php?sta=app)

**Competing interests**: The authors declare that they have no conflict of interest.

## Appendix A   Table of acronyms and symbols

Table A1. Frequently used acronyms and symbols in this paper

| Acronym or symbol | Name or definition |
| --- | --- |
| AERONET | NASA Aerosol Robotic Network |
| AOD | Aerosol optical depth; the vertical integral of aerosol light extinction coefficient |
| APP | Aerosol monitoring sites at Appalachian State University |
| DRE | Aerosol direct radiative effect. In this paper, we only consider clear-sky, diurnally averaged DRE |
| ESRL | National Oceanic and Atmospheric Administration (NOAA) Earth System Research Laboratory |
| SGP | NOAA ESRL's cooperative Southern Great Plains, OK aerosol monitoring site, operated by the U.S. Department of Energy Atmospheric Radiation Measurement (DOE ARM) program |
| g | Scattering asymmetry parameter |
| MODIS | Moderate Resolution Imaging Spectrometer, aboard the polar-orbiting NASA Terra and Aqua satellites |
| MPLNET | NASA Micro-pulsed Lidar Network |
| R | Surface reflectance. For the DRE sensitivity and uncertainty, we refer to the spectrally-averaged surface reflection |
| RE | Aerosol radiative efficiency; equal to DRE divided by AOD at 550 nm |
| RTM | Radiative transfer model |
| RH | Relative humidity |
| SBDART | Santa Barbara DISORT Radiative Transfer model used to calculate DRE |
| SE U.S. | Southeastern United States |
| $S_{AOD}$ | Change in DRE per unit change in AOD[a] |
| $S_g$ | Change in DRE per unit change in g[a] |
| $S_{\omega_0}$ | Change in DRE per unit change in $\omega_0$[a] |
| $S_R$ | Change in DRE per unit change in R[a] |

| | |
|---|---|
| TOA | Top-of-atmosphere. For this paper, TOA refers to 100 km above sea level |
| λ | Wavelength of solar radiation (units: nm) |
| $\omega_0$ | Single-scattering albedo; the fraction of aerosol light extinction due to scattering |

[a] Evaluated at base case values of aerosol properties and surface reflectance (Table 1)

**Acknowledgements**: MODIS MYD09A1 eight-day surface reflectance data product is courtesy of Oak Ridge National Laboratory Distributed Active Archive Center (ORNL DAAC). 2014. MODIS subsetted land products, Collection 5 (http://daac.ornl.gov/MODIS/modis.html; Last accessed May 31, 2017). The authors thank the AERONET science team and NOAA ESRL aerosol group for data processing and instrument support. We thank Appalachian State University personnel Mike Hughes and Dana Greene for their help with electronics and machine shop needs at the APP sites We also thank former Appalachian State University students Chastity Holt and Nicholas Hall for their help in setting up the SBDART software.

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

Table 1. Base case values of aerosol optical properties, spectrally-averaged surface reflectance (R), and vegetation surface cover (SC) coefficients used by SBDART to calculate diurnally averaged DRE for the months DEC, MAR, JUN, and SEP. The vegetation SC coefficients produce the spectral reflectance curves in Figs.3 and are the linear combinations SC = (snow, water, vegetation, sand) that best match monthly averaged spectral reflectance measured by MODIS. The base case aerosol optical properties are the monthly median values (Figs.5).

| Property $p_i$ | MAR | JUN | SEP | DEC |
|---|---|---|---|---|
| AOD (550 nm) | 0.05 | 0.15 | 0.10 | 0.02 |
| $\omega_0$ (550 nm) | 0.91 | 0.95 | 0.95 | 0.91 |
| g (550 nm) | 0.57 | 0.62 | 0.67 | 0.61 |
| $\mathring{A}_{sp}$ | 2.1 | 2.0 | 2.2 | 2.1 |
| $\mathring{A}_{ap}$ | 1.2 | 0.6 | 0.8 | 1.2 |
| $\mathring{A}_{AOD}$ | 1.3 | 1.7 | 1.8 | 1.4 |
| Vegetation SC coefficients | (0,0,0.30,0.20) | (0,0,0,0.75) | (0,0,0.05,0.55) | (0,0,0.40,0.15) |
| R | 0.12 | 0.21 | 0.16 | 0.13 |

5    Table 2. Range over which aerosol optical properties and spectrally-averaged surface reflectance used to calculate DRE sensitivities are varied, along with the measurement uncertainties used in the DRE uncertainty calculations.

| Parameter | Range varied in sensitivity curves | Measurement uncertainty (Source) |
|---|---|---|
| AOD at 550nm | 0.0-0.3 | 0.01 (Eck et al., 1999) |
| $\omega_0$ at 550nm | 0.75-1.0 | 0.02-0.03[a] (Sherman et al., 2015; Titos et al, 2016) |
| g at 550nm | 0.50-0.75 | 0.01[a] (Sherman et al., 2015; Titos et al., 2016) |
| R | 0.0-0.30 | 0.05*R[b] (Vermote and Saleous, 2006) |

[a] Values calculated based on equations S8-S9 of supplement to Sherman et al., 2015, using uncertainties in humidified scattering coefficient reported by Titos et al. (2016). Lower bound of $\Delta \omega_0$ is used for cold-
10    season months ($\omega_0 \sim 0.91$) and upper bound is used for warm season months ($\omega_0 \sim 0.95$)

[b] For uncertainty in R, we use a wavelength-independent $\Delta R \sim 0.02$, which corresponds to reflectance R of 40% and thus represents an upper bound for APP (Fig.3b).

5   Table 3. Sensitivity of top-of-atmosphere (TOA) and surface DRE to AOD, $\omega_0$, g, and R. Sensitivities $S_i$ are calculated as the slope of DRE versus $\rho_i$, curve, evaluated at base case values (Table 1). All sensitivities are in units of W m$^{-2}$ per unit change in the parameter $\rho_i$.  The correlations between aerosol optical properties are used along with uncertainties (Table 2) to calculate the covariances used in the DRE uncertainty calculations (Tables 4 and 5).

| Property $p_i$ | MAR | JUN | SEP | DEC |
|---|---|---|---|---|
| TOA $S_{AOD}$ | -47 | -35 | -34 | -43 |
| TOA $S_{\omega 0}$ | -9.1 | -39 | -18 | -2.6 |
| TOA $S_g$ | 5.9 | 18 | 12 | 1.8 |
| TOA $S_R$ | 7.9 | 17 | 9.1 | 2.0 |
| Surface $S_{AOD}$ | -90 | -69 | -61 | -72 |
| Surface $S_{\omega 0}$ | 16 | 54 | 30 | 4.3 |
| Surface $S_g$ | 6.1 | 19 | 12 | 1.9 |
| Surface $S_R$ | 6.5 | 14 | 7.7 | 1.8 |
| Corr(AOD,$\omega_0$) | -0.02 | 0.25 | 0.57 | 0.10 |
| Corr(AOD,g) | -0.08 | 0.30 | 0.56 | 0.15 |
| Corr($\omega_0$,g) | 0.78 | 0.79 | 0.85 | 0.84 |

5 Table 4. Calculated measurement uncertainties in DRE at the TOA, using the sensitivities and correlations given in Table 3 and measurement uncertainties given in Table 2 as inputs to Eq. (7). Units of $\Delta$DRE are W m$^{-2}$. The uncertainties associated with aerosol optical depth are calculated twice; once using AERONET AOD uncertainties and once using the lower bound for MODIS AOD uncertainty (shown in parentheses).

| | MAR | JUN | SEP | DEC |
|---|---|---|---|---|
| $\Delta$DRE$_{AOD}$ | 0.47 (2.3) | 0.35 (1.8) | 0.34 (1.7) | 0.43 (2.1) |
| $\Delta$DRE$_{\omega 0}$ | 0.27 | 0.77 | 0.36 | 0.079 |
| $\Delta$DRE$_g$ | 0.059 | 0.18 | 0.12 | 0.018 |
| $\Delta$DRE$_R$ | 0.16 | 0.34 | 0.18 | 0.04 |
| Sum of covariance terms | 0.016 (-0.022) | 0.40 (1.1) | 0.25 (0.99) | 0.012 (0.048) |
| $\Delta$DRE (covariance terms not included) | 0.58 (2.3) | 0.97 (1.9) | 0.56 (1.7) | 0.44 (2.1) |
| $\Delta$DRE (covariance terms included) | 0.58 (2.3) | 1.1 (2.2) | 0.74 (2.0) | 0.45 (2.1) |
| DRE[a] (Base case) | -2.4 | -5.7 | -3.6 | -0.91 |
| $\Delta$DRE[a] / DRE (Base Case) | 0.24 (0.97) | 0.20 (0.39) | 0.20 (0.56) | 0.49 (2.4) |

[a] Uncertainty includes covariance terms

Table 5. Calculated measurement uncertainties in DRE at the surface, using the sensitivities and correlations given in Table 3 and measurement uncertainties given in Table 2 as inputs to Eq. (7). Units of $\Delta$DRE are W m$^{-2}$. The uncertainties associated with aerosol optical depth are calculated twice; once using AERONET AOD uncertainties and once using the lower bound for MODIS AOD uncertainties (shown in parentheses).

|  | MAR | JUN | SEP | DEC |
|---|---|---|---|---|
| $\Delta$DRE$_{AOD}$ | 0.90 (4.5) | 0.69 (3.5) | 0.61 (3.0) | 0.72 (3.6) |
| $\Delta$DRE$_{\omega 0}$ | 0.47 | 1.1 | 0.60 | 0.13 |
| $\Delta$DRE$_g$ | 0.061 | 0.19 | 0.12 | 0.019 |
| $\Delta$DRE$_R$ | 0.13 | 0.28 | 0.15 | 0.036 |
| Sum of covariance terms | 0.019(-0.083) | 0.77 (2.6) | 0.62 (2.6) | 0.026 (0.12) |
| $\Delta$DRE (covariance terms not included) | 1.0 (4.5) | 1.3 (3.6) | 0.89 (3.1) | 0.73 (3.6) |
| $\Delta$DRE (covariance terms included) | 1.0 (4.5) | 1.6 (4.0) | 1.2 (3.5) | 0.75 (3.6) |
| DRE (Base case) | -4.6 | -11 | -6.3 | -1.5 |
| $\Delta$DRE[1] / DRE (Base Case) | 0.22 (0.98) | 0.15 (0.37) | 0.19 (0.56) | 0.50 (2.4) |

[1]Uncertainty includes covariance terms

Table 6. Calculated uncertainties in DRE at the TOA and at the surface due to diurnal aerosol variability, using the sensitivities and correlations given in Table 3 and estimates of aerosol diurnal variability as inputs to Eq. (7). Units of $\Delta$DRE are W m$^{-2}$. Uncertainties due to diurnal variability in surface albedo are not included in the calculation.

| | MAR | JUN | SEP | DEC |
|---|---|---|---|---|
| $\Delta$AOD | 0.01 | 0.02 | 0.02 | 0.01 |
| $\Delta\omega_0$ | 0.01 | 0.01 | 0.01 | 0.01 |
| $\Delta$g | 0.02 | 0.03 | 0.03 | 0.01 |
| TOA $\Delta$DRE$_{AOD}$ | 0.47 | 0.70 | 0.69 | 0.43 |
| TOA $\Delta$DRE$_{\omega0}$ | 0.091 | 0.39 | 0.18 | 0.026 |
| TOA $\Delta$DRE$_g$ | 0.12 | 0.54 | 0.35 | 0.018 |
| TOA $\Delta$DRE | 0.50 | 1.3 | 1.1 | 0.43 |
| TOA $\Delta$DRE[1] / DRE (Base Case) | 0.20 | 0.22 | 0.29 | 0.48 |
| Surface $\Delta$DRE$_{AOD}$ | 0.90 | 1.4 | 1.2 | 0.72 |
| Surface $\Delta$DRE$_{\omega0}$ | 0.16 | 0.54 | 0.30 | 0.04 |
| Surface $\Delta$DRE$_g$ | 0.12 | 0.57 | 0.36 | 0.019 |
| Surface $\Delta$DRE | 0.92 | 2.0 | 1.7 | 0.73 |
| Surface $\Delta$DRE[1] / DRE (Base Case) | 0.20 | 0.18 | 0.27 | 0.49 |

[1]Uncertainty includes covariance terms

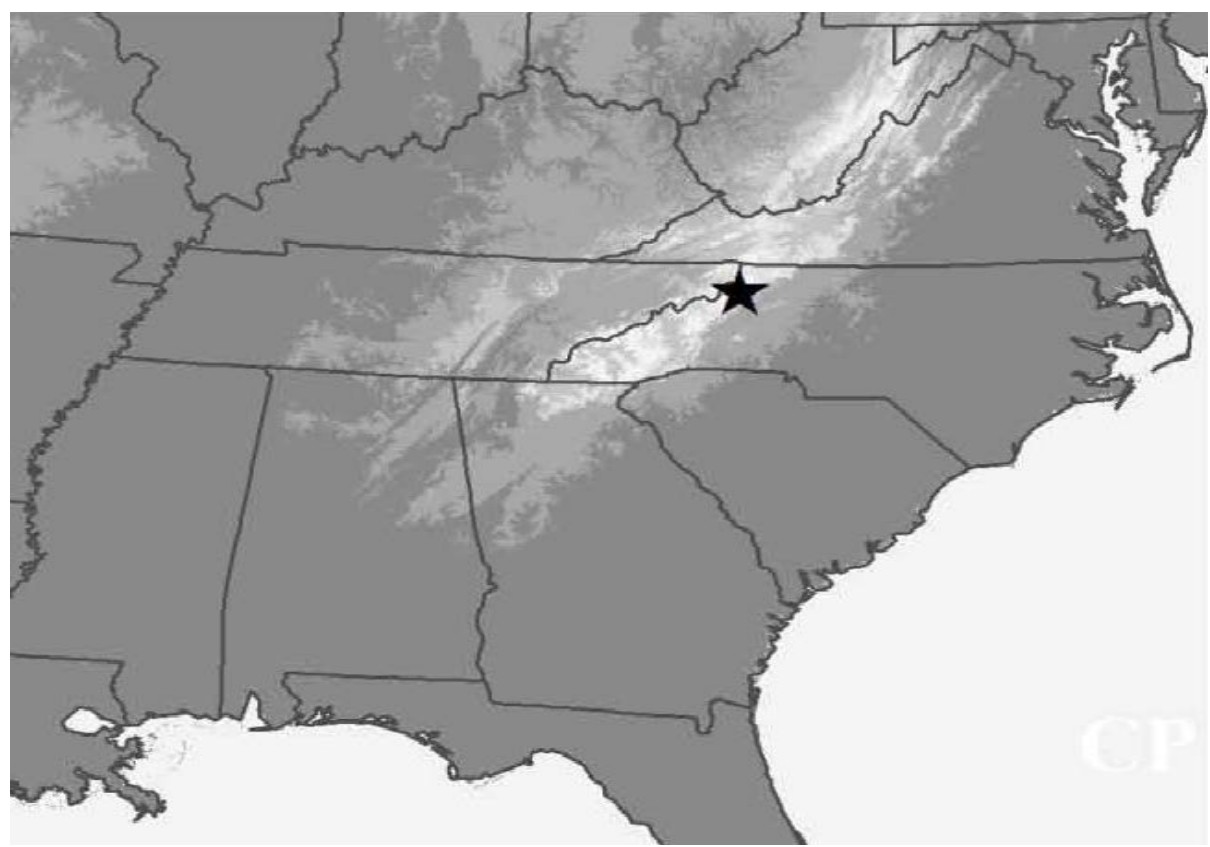

Figure 1. Map of the SE U.S. showing the location of Appalachian State University (APP) in Boone, NC
(36.21$^0$ N, 81.69$^0$ W, 1080 m above sea level).  White shades denote mountain elevations.

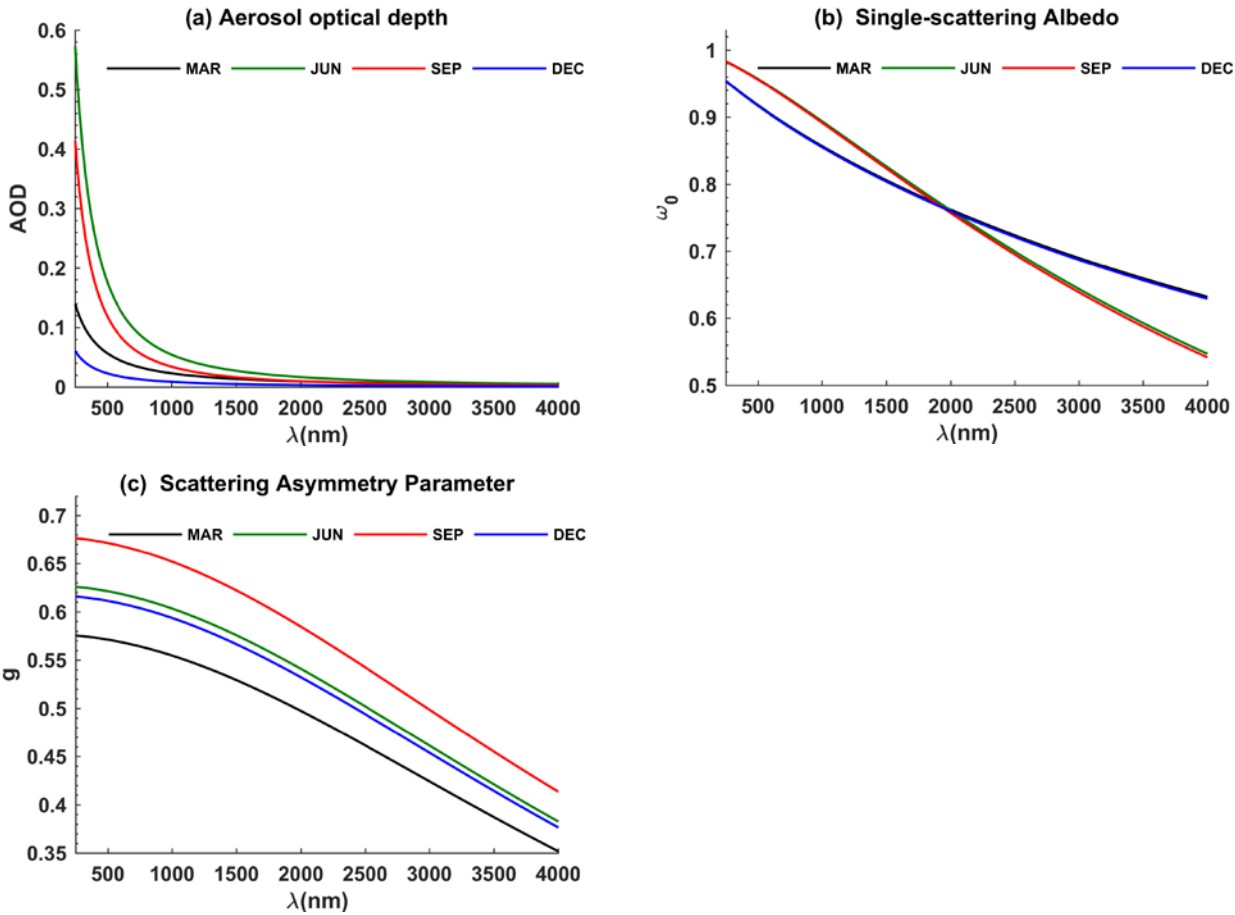

Figure 2. Wavelength dependence of AOD, $\omega_0$, and g, calculated using Eq(s). 1 thru 3 and base case aerosol optical properties for each season (Table 1). The MAR and DEC curves are nearly identical in (b), as are the JUN and SEP curves in (b), due to the use of nearly identical base case aerosol optical properties in generating the curves.

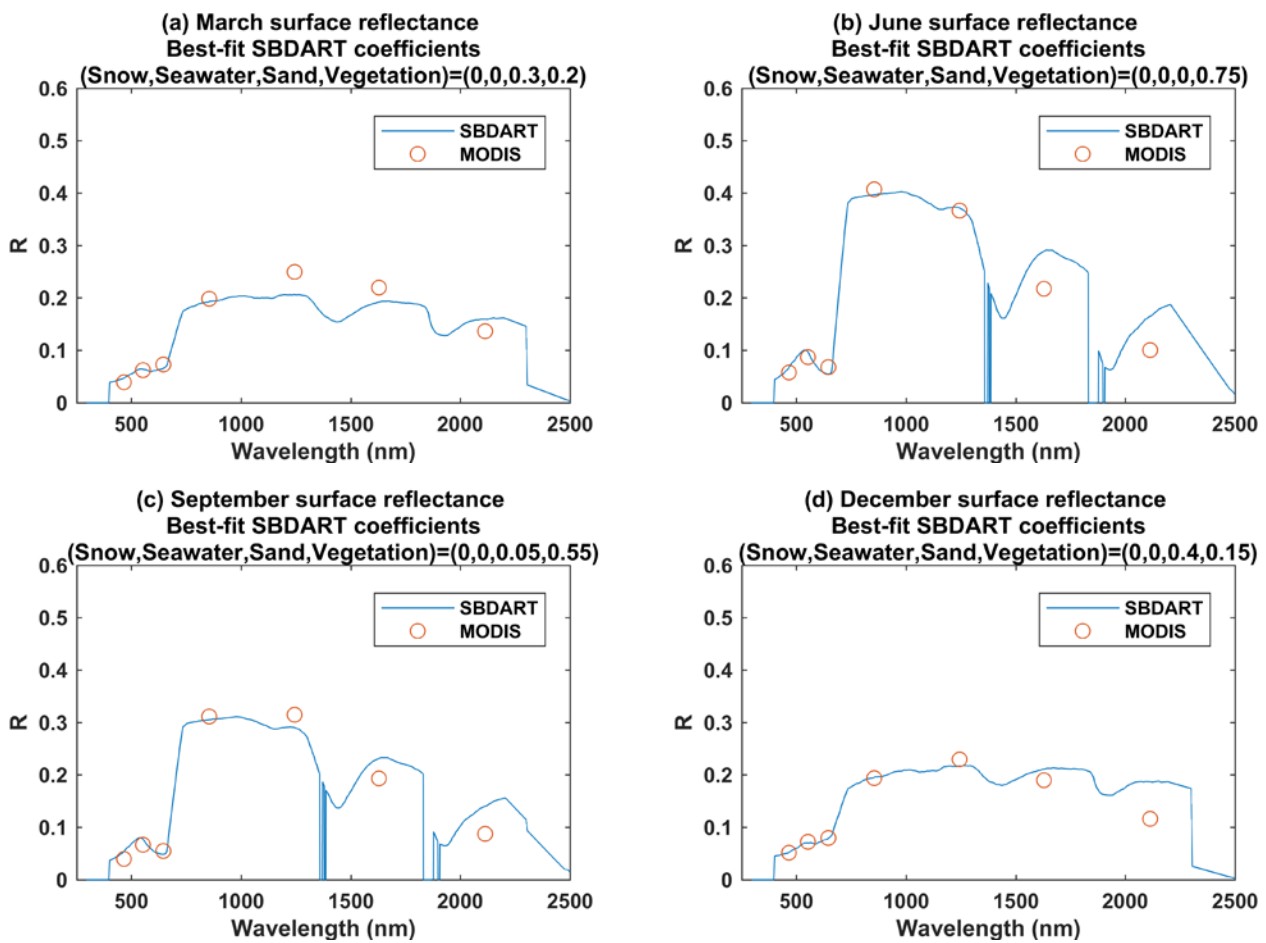

Figure 3. Spectral surface reflectance over the wavelength range 250 to 2500 nm for (a) March; (b) June; (c) September; and (d) December. The MODIS values for each month represents average of MODIS Aqua spectral surface reflectance values for that month, over the study period. The SBDART curve is based on spectral reflectance produced by the vegetation types whose combination provided best fit with MODIS-measured spectral reflectance.

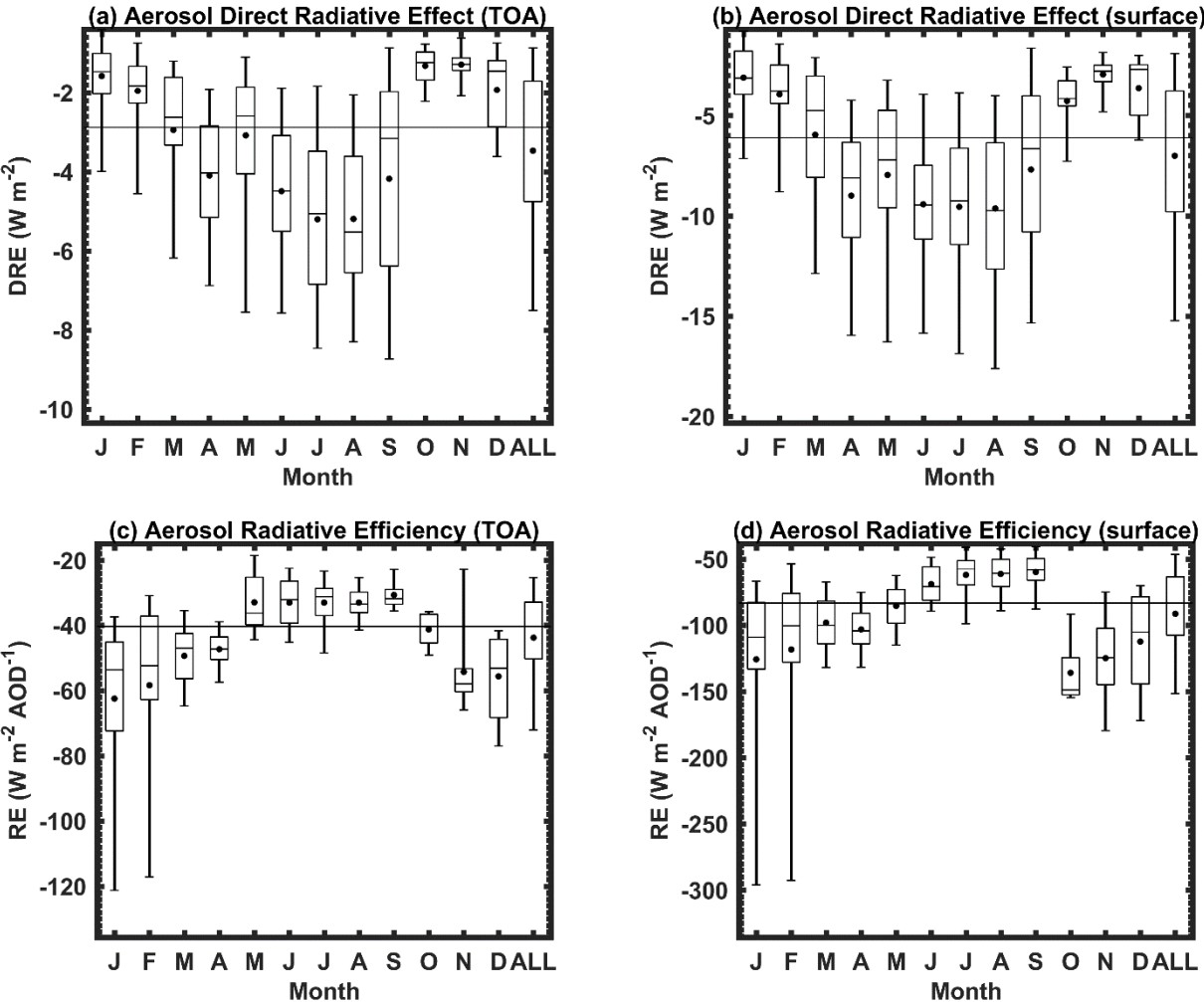

Figure 4. Boxplots of calculated monthly binned aerosol DRE and RE at the top of atmosphere and the surface. The 'ALL' box provides the statistics for all days in June 2012 thru February 2016 period. The mean for each month is denoted by the dot while the horizontal bar represents the median. The top and bottom of the box represent 75th and 25th percentiles while the top and bottom whisker extend to the 95th and 5th percentiles, respectively. The horizontal line drawn through all boxes of each plot represents the median value over the entire period (all months).

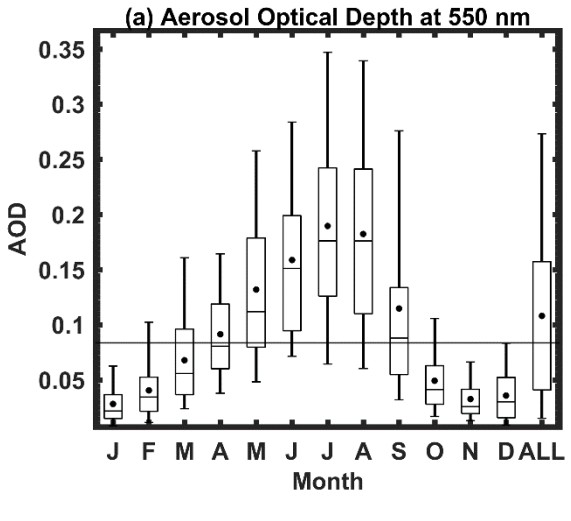

**(a) Aerosol Optical Depth at 550 nm**

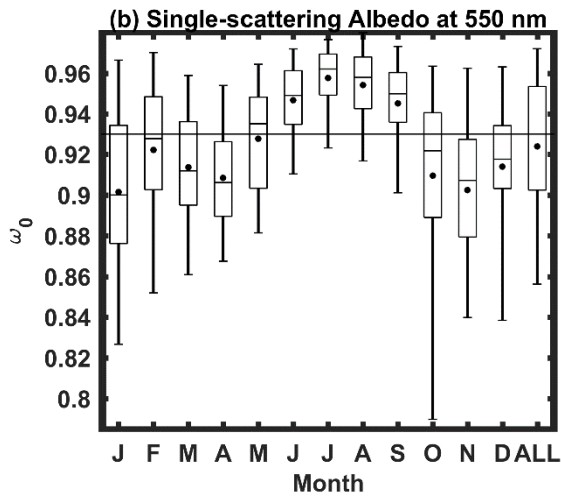

**(b) Single-scattering Albedo at 550 nm**

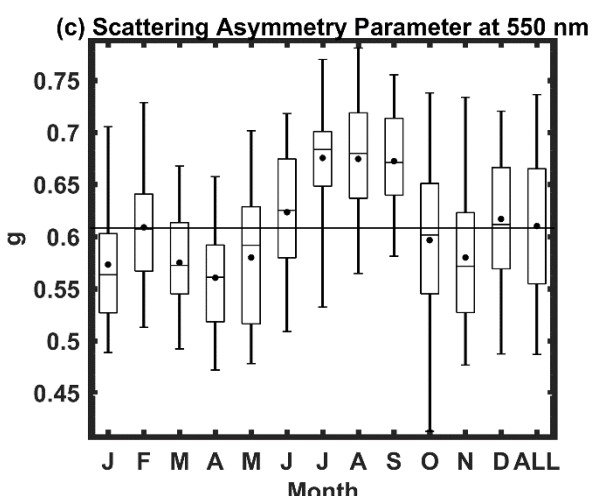

**(c) Scattering Asymmetry Parameter at 550 nm**

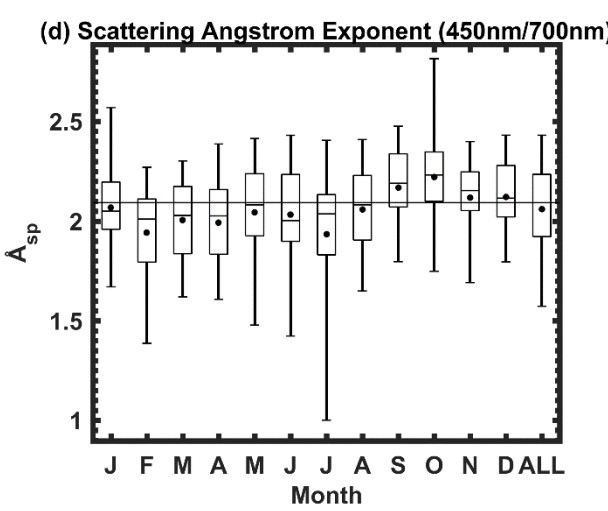

**(d) Scattering Angstrom Exponent (450nm/700nm)**

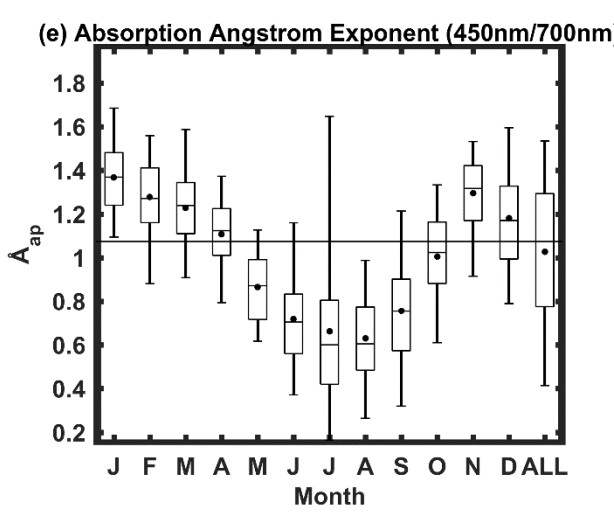

**(e) Absorption Angstrom Exponent (450nm/700nm)**

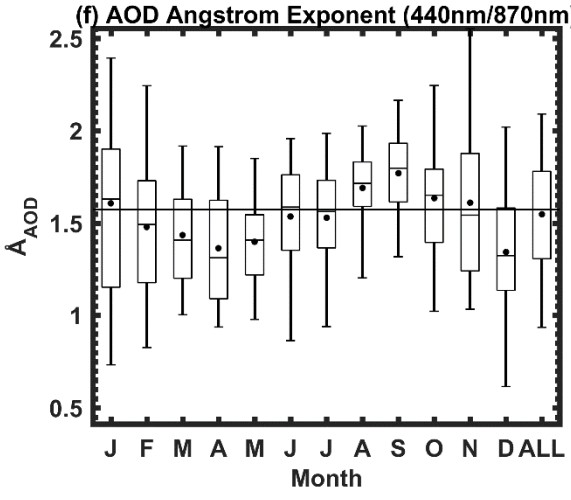

**(f) AOD Angstrom Exponent (440nm/870nm)**

Figure 5. Boxplots of monthly binned aerosol optical properties at APP. The 'ALL' box provides the statistics for all days in June 2012 thru February 2016 period. The mean for each month is denoted by the dot while the horizontal bar represents the median. The top and bottom of the box represent 75th and 25th percentiles while the top and bottom whisker extend to the 95th and 5th percentiles, respectively. The horizontal line drawn through all boxes of each plot represents the median value over the entire period (all months).

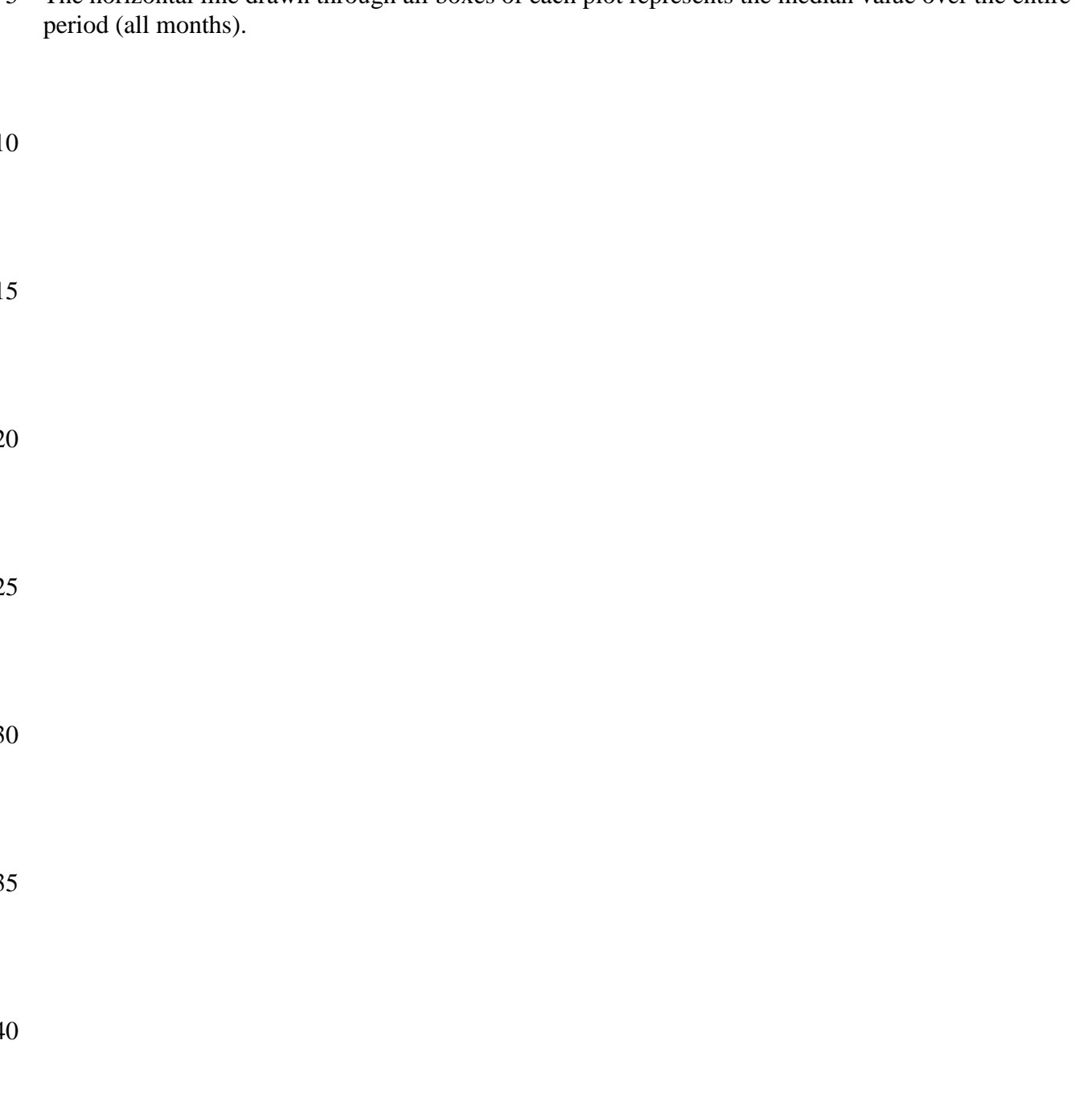

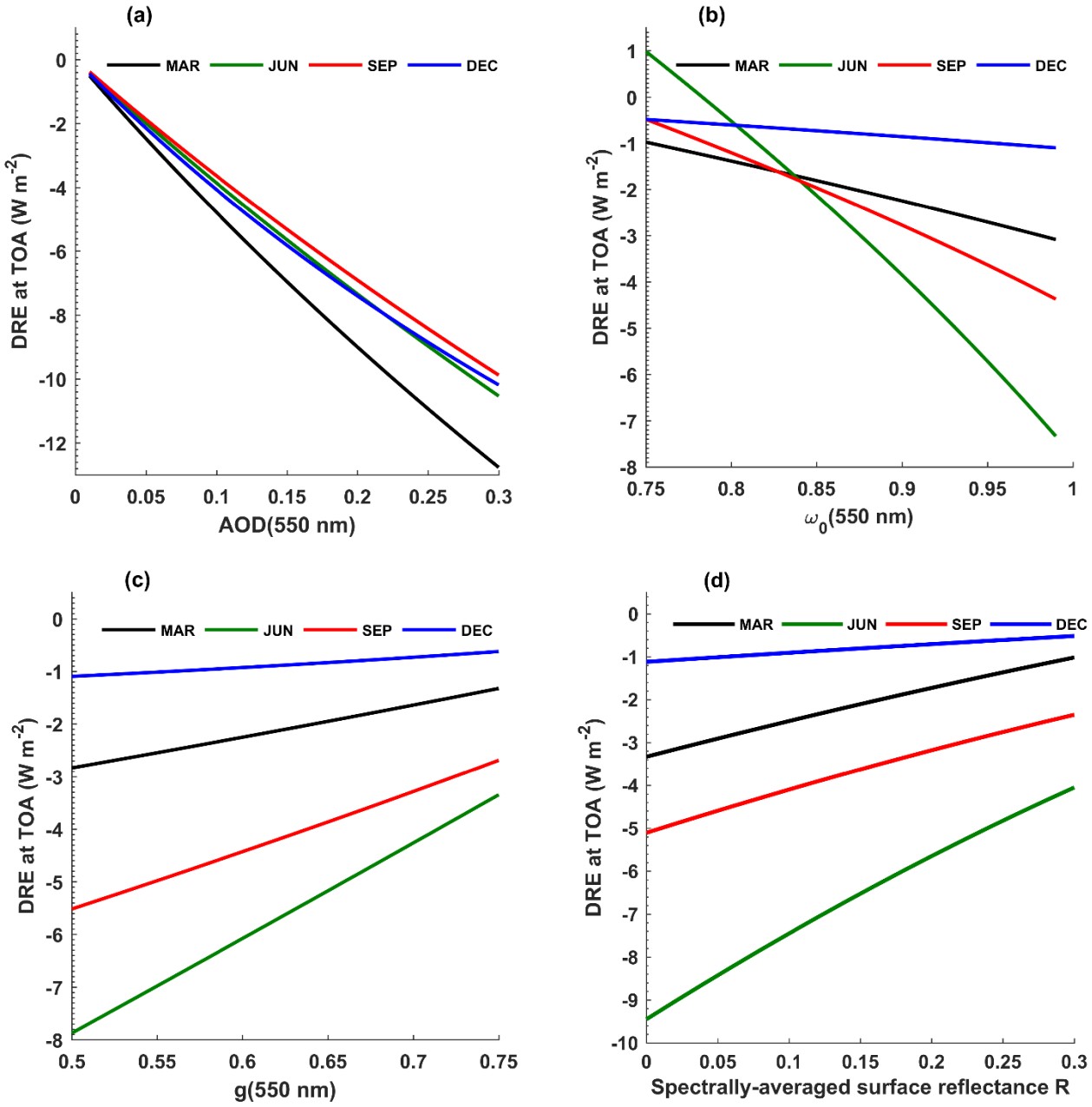

Figure 6. Seasonal dependence of the sensitivity of top-of-atmosphere (TOA) aerosol DRE to (a) AOD; (b) $\omega_0$; (c) g; and (d) spectrally-averaged surface reflectance R

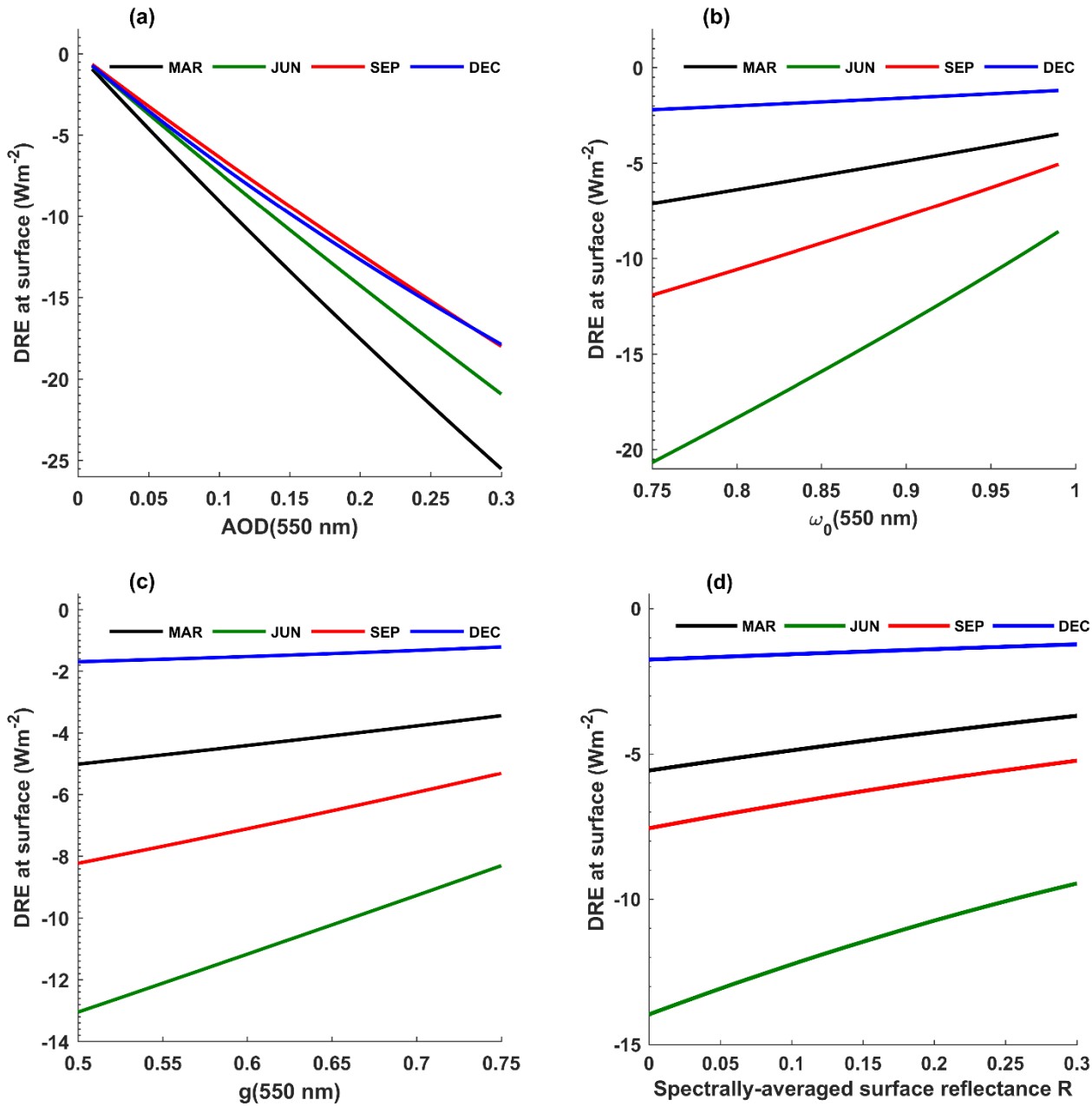

Figure 7. Seasonal dependence of the sensitivity of aerosol DRE at the surface to (a) AOD; (b) $\omega_0$; (c) g; and (d) spectrally-averaged surface reflectance R.

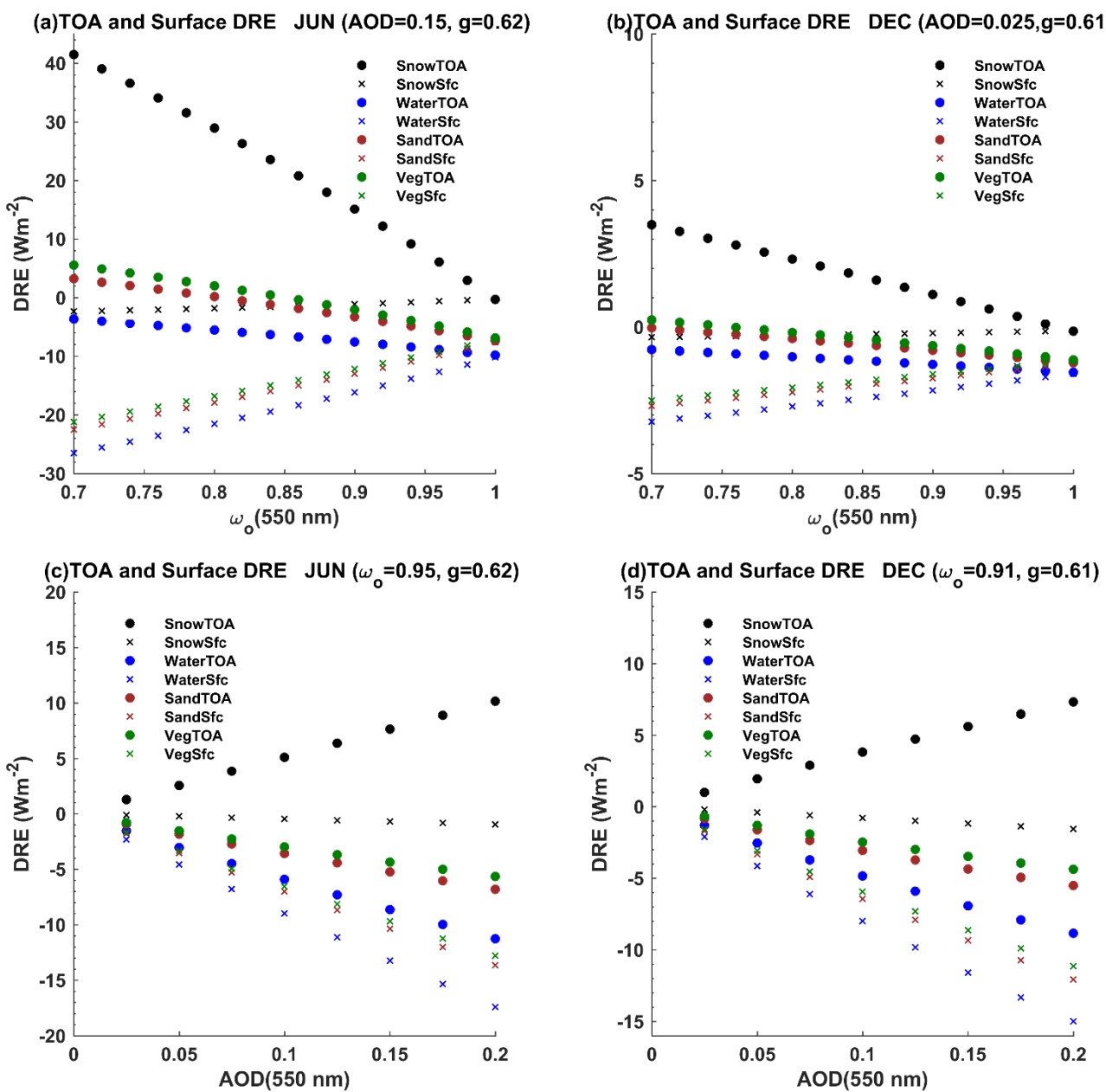

Figure 8. TOA and surface DRE versus $\omega_o$ and AOD for JUN and DEC, for each of the four surface types (snow, seawater, sand, and vegetation) used by the SBDART RTM. The base-case values for the fixed aerosol properties (Table 1) are listed on the plot titles.

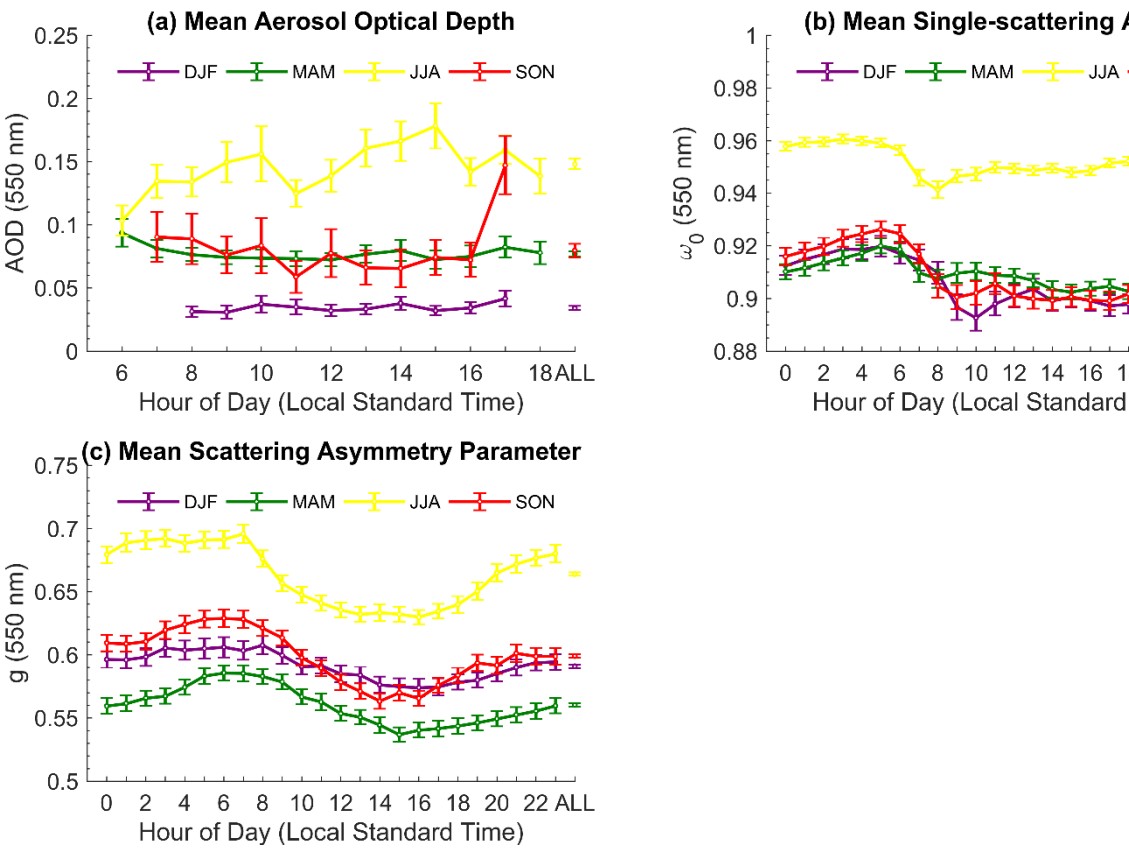

Figure 9. Diurnal cycles of mean (a) AOD; (b) $\omega_0$; and (c) g at 550 nm for winter (DJF), spring (MAM), summer (JJA), and autumn (SON). The 'ALL' points are the mean values over all hours of day. Standard errors of the mean values are plotted as error bars.