# Peer review of "Measurement-based climatology of aerosol direct radiative effect, its sensitivities, and uncertainties from a background southeast U.S. site"

_Atmospheric Chemistry and Physics, 2017_

## Referee Comment (RC1) · Anonymous Referee #1 · 17 Jul 2017

**1 General Comments:**

Authors have analysed about four years of ground-based measurements of aerosol optical depth (AOD), scattering coefficient and absorption coefficient data at a remote site in south-east U.S. Measured daily mean values are used in radiative transfer code to estimate aerosol direct radiative effect (DRE) and radiative efficiency (RE; DRE per unit AOD at 550 nm). They also present analysis of uncertainties in the estimated DRE and RE values. In spite of lot of progress, aerosol radiative effects remains a big

source of uncertainty in projecting future climate change. The RE numbers are often used directly in models to simulate effect of aerosol on atmospheric dynamics as well as indirectly to validate model for their ability to simulate aerosol concentrations and their effects on climate. In this context, authors' contribution is important.

Overall manuscript is well written. Introduction provides concisely and clearly the importance of DRE and objectives of the manuscript. Methodology is described with sufficiently detail to allow others to reproduce their results and the results are provided with good clarity describing how their finding improve upon previous study.

As a suggestion for improvement. The manuscript focuses a great deal on sensitivities to AOD, single scattering albedo (SSA), asymmetry parameter (g) and surface reflectance (R). I believe sensitivities are intrinsic property of radiative transfer model and their emphasis would have made more sense if the results were about comparison of different radiative transfer models. In the present case, their findings on sensitivities will differ from others (previous studies) only to extent differences in base cases and impact of non-linearity over the range of difference. I believe emphasis should have been more on seasonal variations in aerosol properties and how they differ from generic aerosol models used in various models and satellite retrieval algorithms, and ultimately what would be the penalty in terms of error in DRE if the generic models are used instead of measurements.

**2 Technical comment:**

1. Authors discuss effect of measurement uncertainties on uncertainties in DRE. However, it is rare that DRE is estimated for instantaneous values measured by various instruments. Generally, required parameters are averaged over certain time-period (typically one day) and will have associated variabilities, quite often larger than instrumental error leading to further uncertainty in DRE estimation. I

am not clear about whether authors mention uncertainty in DRE including variability of input parameters or only of measurement error. Authors may consider including discussion on uncertainty that arises from variability of the input parameters in addition to the measurement error.

2. Authors have used power law equation to extrapolate AOD and SSA beyond visible wavelength. Originally, the power law was derived for visible wavelength range and there aren't many evidences to suggest applicability of the law in infrared. At the same time, I believe authors may not have made big error in DRE numbers in doing so as the solar energy in that part of spectrum is very little compared to visible range. However, I feel a caveat in the manuscript is necessary to reflect that power law assumption may or may not be valid in infra-red region of the spectrum.

3. Authors imply on page 10(line 2 to 4) that uncertainty in SSA at higher RH is not known. However in the section 3, authors have mentioned that the site is equipped with scanning humidograph to study effect of RH on scattering and absorption coefficient. Authors may explain why can't this data be used to find uncertainty in SSA at high RH?

4. Authors present sensitivity of DRE to surface reflectance ($S_R$) at TOA and surface as $3.3Wm^{-2}$ and $2.7Wm^{-2}$ during June and $0.22Wm^{-2}$ and $0.20Wm^{-2}$ during December. What surprises me is the very small difference in $S_R$ values at TOA and at surface. It is common knowledge that surface reflectivity will have very little effect on DRE at surface but can have significant effect at TOA. One can read reason for it in Chung (2012). In other words, a significant difference is expected between TOA and surface $S_R$ values. See for example Figure 10 of Gadhavi and Jayaraman (2004) who have used similar approach and the same radiative transfer code (SBDART) to calculate DRE (they called it radiative forcing) as a function of AOD and surface reflectance. They have reported that when surface

type changes from sea to sand (which is large change in surface reflectance) it causes a little change in DRE at surface but a large change in DRE at TOA for a fixed AOD. The values reported in the current manuscript may not be wrong but a thorough discussion needs to be included why their finding is at variance with others or the common knowledge. I believe such a discussion will add value to their manuscript as it will lead to better understanding of how non-aerosol parameter affects aerosol radiative forcing.

**3  Typing Errors:**

Page 35 caption of Fig. 1: Longitude number of the site should have suffix "W".

**4  References:**

1. Chul Eddy Chung (2012). Aerosol Direct Radiative Forcing: A Review, Atmospheric Aerosols - Regional Characteristics - Chemistry and Physics, Hayder Abdul-Razzak (Ed.), InTech, DOI: 10.5772/50248. Available from: https://www.intechopen.com/books/atmospheric-aerosols-regional-characteristics-chemistry-and-physics/aerosol-direct-radiative-forcing-a-review

2. Gadhavi, H. and Jayaraman (2004), A. Aerosol characteristics and aerosol radiative forcing over Maitri, Antarctica, Current Science, 86, 296-304.

---

## Referee Comment (RC2) · Anonymous Referee #2 · 30 Nov 2017

In this study, the authors have analyzed 4 years of aerosol properties, estimated DRE and examined the sensitivity of DRE to key parameters. The paper is very well written with clear context of the present work in view of the published works. Methodology is clearly defined and the DRE estimates are discussed in view of the uncertainty. It is a good contribution to the growing literature on aerosol-climate interaction.

Few minor points are required to be clarified though.

1. Aerosol properties are retrieved during daytime in presence of solar radiation. How

are then DRE estimated for 24 hours? Or is it estimated for a range of SZA?

2. Sec 4.4: what does rho with subscript 'j' represent? Is it another aerosol parameter?

3. How closely do the SBDART aerosol profile and MPLNET profile match?

4. Page 22, lines 18-19: mention the range for 'moderate AOD' to have a perspective, similarly for 'low AOD'.

---

## Author Comment (AC2) · 11 Jan 2018

We thank anonymous reviewer #2 for her/his excellent suggestions, which we hope will lead to improved paper readability. We've gone to great lengths to implement nearly all the suggestions made by both reviewers and believe that these changes have significantly improved the paper. We structure our responses to each reviewer comment/suggestion as follows: (1) Reviewer 2 Comment xx, where xx is the comment number; (2) Authors' response; and (3) Changes to Paper.

[Figure]

Reviewer 2 Comment 1: Aerosol properties are retrieved during daytime in presence of solar radiation. How are then DRE estimated for 24 hours? Or is it estimated for a range of SZA?

Authors' Response: To estimate diurnally-averaged DRE, we apply the daily-averaged aerosol optical properties as inputs to the RTM for each of the 24 hours, as described in the first paragraph of Sect. 4.2. Using daily-averaged aerosol properties as inputs to the RTM for each of the 24 hours basically amounts to integrating over the range of SZA, so that the effect of SZA on diurnally-averaged DRE is averaged out. The in situ aerosol measurements used by the radiative transfer model ($\omega 0$ and g) as part of NOAA ESRL are retrieved over all 24 hours so the 'daily-averaged' $\omega 0$ and g represent true 24-hour averages. Aerosol optical depth (AOD) measured as part of NASA AERONET requires sunlight and is only measured during presence of solar radiation (i.e. daylight hours), as the reviewer points out. Our 'daily-averaged' AOD is thus calculated based only on these daytime values and may or may not be representative of AOD during nighttime hours. However, AOD during night-time hours does not affect the calculations of the shortwave solar fluxes, since these shortwave fluxes (both with and without aerosols turned on in the RTM) are zero during nighttime, leading to DRE=0 for these hours.

Changes to Paper: We clarify these points by modifying the first paragraph of Sect. 4.2 so that it now reads as follows. We embolden the additions/modifications to the paragraph: "For the study of seasonal DRE variability (Sect. 5.1), we use the SBDART model to calculate diurnally averaged DRE at the TOA and at the surface, for 418 days during the period 14 June 2012 thru 28 June 2016. We then bin the DRE by month (Figs. 4a and 4b). For each of the 418 days, we calculate DRE for each hour to account for the effect of varying solar geometry on the calculation of diurnally-averaged DRE. For each hour, we supply daily-averaged AOD($\lambda$), $\omega 0(\lambda$), and g($\lambda$), along with monthly averaged spectral surface reflectance (R), as inputs to the SBDART model. Upwelling and downwelling broadband shortwave fluxes for that hour are calculated with average

measured aerosol properties and then with no aerosols and their difference is used to calculate DRE using Eq. (5). The process is repeated for all 24 hours and the results averaged to yield diurnally averaged DRE. Since AOD is only measured during daytime hours, the daily-averaged AOD used as RTM input may or may not be representative of AOD during night-time hours. However, AOD during night-time hours does not affect the calculation of shortwave solar fluxes, since these fluxes (both with and without aerosols) are zero during night-time (leading to calculated DRE=0 for these hours)."

Reviewer 2 Comment 2: Sec 4.4: what does rho with subscript 'j' represent? Is it another aerosol parameter?

Authors' Response: No. The equation (Eq.6) used to calculate DRE uncertainties due to uncertainties in AOD, $\omega 0$, g, and R is first written as a summation over the four individual uncertainties, before being explicitly spelled out in Eq.7.

Changes to Paper: We clarified the use of the subscripts with the following sentence, after Eq.6: "The double summation 'i' and 'j' is over the four RTM input parameters (AOD, $\omega 0$, g, and R)."

Reviewer 2 Comment 3: How closely do the SBDART aerosol profile and MPLNET profile match?

Authors' Response: Since are APP site was not added to MPLNET until March 2016 (after the period of the current study), our lidar-measured vertical aerosol profiles are not quality-assured and therefore not used in the current study, other than qualitative inspection to verify that aerosols are largely confined to the lowest 1 to 2 km of atmosphere above APP (first paragraph of Sect.3.1). We state in the first paragraph of Sect. 3.1.2 that "Most vertical profiles of aerosol normalized relative backscatter measured by the lidar at APP during part of the study period and afterward (as part of MPLNET) show a qualitatively exponential decay with height and an absence of aerosol layers aloft (unpublished result)" and state in the final paragraph of Sect. 4.1 that "Most vertical profiles of aerosol normalized relative backscatter measured by the lidar at APP

during part of the study period and afterward (as part of MPLNET) show a qualitatively exponential decay with height and an absence of aerosol layers aloft (unpublished result)". These assertions are based on visual inspections of the lidar-measured normalized relative backscatter (NRB) vertical profiles. Most of the NRB profiles decay relatively smoothly with increasing altitude (quasi-exponentially), with NRB dropping to ~1/3 of the peak values at altitudes between 1 and 2 km (more often than not below 1.5km). This decay is similar to the vertical dependence assumed by the standard SBDART vertical profiles used in the study, which treat the aerosol density vertical distribution as exponentially decaying, with scale heights between 1.05-1.51km. The scale heights used by SBDART are calculated from the near-surface aerosol extinction coefficients, which we supply to SBDART. Although vertical distribution of aerosols is believed to be a second-order effect in the calculation of aerosol DRE for primarily scattering aerosols (McComiskey et al., 2008), we plan to study its influence on DRE at APP as part of a future publication. However, MPLNET is currently upgrading their processing to Version 3 and quantitative, quality-assured aerosol profiles from the APP MPLNET site are not yet available for download.

Changes to Paper: We further clarified the final paragraph of Sect. 4.1 to read as follows, with the additions emboldened: "Vertical distribution of aerosols is believed to be a second-order effect in the calculation of aerosol DRE for primarily scattering aerosols (McComiskey et al., 2008) and we use the SBDART default vertical aerosol density profile in this initial study. The default profile uses an assumed exponential decrease in aerosol density with a scale height inversely proportional to surface-level aerosol light extinction coefficient at 550 nm (Ricchiazzi et al., 1998), which is calculated as the sum of the measured $\sigma$sp and $\sigma$ap (Sect. 3.1.2). The overall curve is scaled by the AOD (Sect.3.1.1). Aerosol density scale heights used by SBDART range from 1.05 to 1.51 km, which qualitatively agree with typical MPL-measured normalized relative backscatter profiles under clear sky conditions at APP (Sect. 2)."

Reviewer 2 Comment 4: Page 22, lines 18-19: mention the range for 'moderate AOD'

to have a perspective, similarly for 'low AOD'.

Authors' Response: Done

Changes to Paper: We have clarified the passage mentioned by the reviewer as follows: "Unlike the McComiskey et al.(2008) study, we include the effect of covariances amongst aerosol optical properties in order to determine their effect on DRE uncertainty. Covarience impacts on DRE uncertainty at APP are negligible for low AOD conditions (AOD≤0.05 at 550nm) during winter and surrounding months but do increase $\Delta$DRE by ~0.2 to 0.3 Wm-2 under moderate and high AOD conditions (AOD≥0.10 at 550nm) during summer and surrounding months." We also qualify 'low AOD', 'moderate AOD', and 'high AOD' when they are used in the other sections of the paper.

Please also note the supplement to this comment:
https://www.atmos-chem-phys-discuss.net/acp-2017-513/acp-2017-513-AC2-supplement.pdf

---

## Author Comment (AC3) · 11 Jan 2018

We thank anonymous reviewer #1 for her/his excellent suggestions, particularly those related to the Analysis/Discussion section. We've gone to great lengths to implement nearly all of the suggestions made by both reviewers and believe that these changes have significantly improved the paper. We structure our responses to each reviewer comment/suggestion as follows: (1) Reviewer 2 Comment xx, where xx is the comment number; (2) Authors' response; and (3) Changes to Paper.

[Figure]

Reviewer 1 Comment 1: As a suggestion for improvement. The manuscript focuses a great deal on sensitivities to AOD, single scattering albedo (SSA), asymmetry parameter (g) and surface reflectance (R). I believe sensitivities are intrinsic property of radiative transfer model and their emphasis would have made more sense if the results were about comparison of different radiative transfer models. In the present case, their findings on sensitivities will differ from others (previous studies) only to extent differences in base cases and impact of non-linearity over the range of difference. I believe emphasis should have been more on seasonal variations in aerosol properties and how they differ from generic aerosol models used in various models and satellite retrieval algorithms, and ultimately what would be the penalty in terms of error in DRE if the generic models are used instead of measurements.

Authors' Response: We respectfully disagree with the reviewer's assertion that the sensitivities are intrinsic property of the RTM (if we are correctly interpreting her/him) and point to the results from two studies. As part of a radiative transfer closure study, Michalsky et al.(2006) found that six radiative transfer models (RTMs) were all able to simulate clear-sky direct and diffuse shortwave fluxes to within 1.0% and 1.9%, respectively, of the measured fluxes, provided that all models used the same aerosol inputs. They concluded that the largest source of difference in the RTM outputs is likely due to how the RTM extrapolates the aerosol optical properties used as inputs (particularly AOD) to unspecified wavelengths. As a follow-up to this study, McComiskey et al. (2008) showed that the sensitivities of clear-sky DRE to changes in aerosol inputs were not dependent on the model used. Both studies demonstrate that the RTMs are capable of calculating clear-sky DRE with high precision and that DRE uncertainty arises largely from incorrectly-specified aerosol optical properties

The reviewer is likely correct in her/his assessment that the sensitivities are primarily dependent on base-case aerosol optical properties (and not on the model used) and this in fact is why regionally-representative aerosol measurements possessing low uncertainties (such as from NOAA-ESRL and AERONET sites) are needed to improve

DRE estimates. Fortunately, studies (Sherman et al., 2015; Delene and Ogren 2002; and others) have shown that intensive aerosol optical properties such as SSA are not too different between many North American regions and AOD has decreased significantly over much of North America in the past 2 decades. As a result, the results in this paper should be applicable over at least the SE US and likely much of eastern continental North American. The technique can also be easily extended to industrial regions where the sensitivity values may not be applicable, given a co-located NOAA-ESRL/AERONET site (ex: Bondville, IL; Egbert, Ontario; etc).

Changes to Paper: We have added the following text to the first paragraph of the Introduction section: "As part of a recent radiative transfer closure study, Michalsky et al.(2006) found that six radiative transfer models (RTMs) were all able to simulate the observed clear-sky direct and diffuse shortwave fluxes to within 1.0% and 1.9%, respectively, of the measured fluxes, provided that all models used the same aerosol inputs. They concluded that the largest source of difference in the RTM-derived fluxes is likely due to how the RTM extrapolates the aerosol optical properties used as inputs (particularly AOD) to unspecified wavelengths. As a follow-up to this study, Mc-Comiskey et al. (2008) showed that the sensitivities of clear-sky DRE to changes in aerosol inputs was not dependent on the model used. Both studies demonstrate that the RTMs are capable of calculating clear-sky DRE with high precision and that DRE uncertainty arises largely from incorrectly-specified aerosol optical properties, which can result from lack of regionally-representative aerosol measurements, measurement uncertainties, and spatio-temporal aerosol variability."

Reviewer 1 Technical Comment 1: Authors discuss effect of measurement uncertainties on uncertainties in DRE. However, it is rare that DRE is estimated for instantaneous values measured by various instruments. Generally, required parameters are averaged over certain time-period (typically one day) and will have associated variabilities, quite often larger than instrumental error leading to further uncertainty in DRE estimation. I am not clear about whether authors mention uncertainty in DRE including variability of

input parameters or only of measurement error. Authors may consider including discussion on uncertainty that arises from variability of the input parameters in addition to the measurement error.

Authors' Response: The reviewer brings up a very good point, namely that diurnal variability in the aerosol optical properties serving as inputs to the radiative transfer model (AOD, SSA, g) can often lead to DRE uncertainties that are at least as large as DRE uncertainties due to measurement uncertainties. While the main purpose of our paper is to quantify uncertainties in DRE due to the aerosol measurement uncertainties (which likely leads to a lower bound in DRE uncertainty), we do agree that some discussion of diurnal aerosol variability and its effect on DRE calculations is warranted.

Changes to Paper: We have added a short section (Sect. 5.4) to the manuscript, discussing DRE uncertainties due to diurnal aerosol variability. We apply the DRE sensitivity parameters (Sect. 5.2) along with an estimate of aerosol diurnal variability, to estimate diurnally-averaged DRE uncertainties due to diurnal aerosol variability. To estimate diurnal aerosol variability, we apply the method used by Sherman et al (2015) and Sherman et al. (2016), both of which are referenced in the manuscript. For each season, we form hourly averages of all AOD, SSA, and g values at 550nm. We then bin the values by hour of day and form statistics for each hour of the day (mean, standard error of the mean). We also form statistics using all hours of the day (i.e. the entire dataset for that season). We include a new figure (Figs. 8(a)-(c)), containing plots of the diurnal cycle of mean AOD, SSA, and g at 550nm for each season. We include error bars for each hour to indicate confidence in the mean values (i.e. standard error of the mean) and to assess whether the diurnal variability in mean aerosol properties is statistically-significant. We estimate "diurnal variability" of each aerosol input (AOD, SSA, and g), using the difference between the diurnally-averaged values and the mean values for individual hours of the day. For example, suppose that the daily-mean SSA at 550nm during summer is 0.96 and that the mean SSA values for individual hours of the day ranged from 0.94 to 0.98, we would estimate the peak error in using the daily-

averaged SSA (0.96) as △SSA=0.02. Use of the peak error leads to upper bounds on the resulting DRE uncertainty estimates but represent a simple application of the DRE sensitivity parameters to estimate DRE uncertainties. We report the DRE uncertainties due to diurnal aerosol variability in a newly-created table (Table 6). We also preface the phrase "DRE uncertainties" with the word "measurement" throughout the paper, in cases where confusion may exist.

Reviewer 1 Technical Comment 2: Authors have used power law equation to extrapolate AOD and SSA beyond visible wavelength. Originally, the power law was derived for visible wavelength range and there aren't many evidences to suggest applicability of the law in infrared. At the same time, I believe authors may not have made big error in DRE numbers in doing so as the solar energy in that part of spectrum is very little compared to visible range. However, I feel a caveat in the manuscript is necessary to reflect that power law assumption may or may not be valid in infra-red region of the spectrum.

Authors' Response: We agree completely with the reviewer and this comment is supported by the Michalsky et al., 2006 study (See response to Comment 1 above). We have added a caveat to this extent.

Changes to Paper: We have added the following passage to the first paragraph of Sect. 3.1-Aerosol Optical Properties: "We note that the power-law expressions (Eqs. 1,2, and 4) used to extrapolate aerosol properties measured largely at visible wavelengths to the infra-red may or may not represent their true spectral dependence. However, the solar flux in the infra-red is much less than that in the visible so simple aerosol spectral parameterizations should be sufficient for broadband DRE calculations."

Reviewer 1 Technical Comment 3: Authors imply on page 10(line 2 to 4) that uncertainty in SSA at higher RH is not known. However in the section 3, authors have mentioned that the site is equipped with scanning humidograph to study effect of RH on scattering and absorption coefficient. Authors may explain why can't this data be

used to find uncertainty in SSA at high RH?

Authors' Response: A scanning humidograph (Sheridan, et al., 2001) is employed at APP to measure the RH dependence of scattering and hemispheric backscatter coefficients ( $\sigma$sp and $\sigma$bsp) but not absorption coefficient. Radiative transfer models typically only treat the scattering dependence of RH, and assume that absorption changes negligibly with RH. While this approach may or may not hold true for all aerosol types (ex: some organics, sulfur-coated soot), the dependence of absorption on RH is experimentally very difficult for all but laboratory studies (especially at high RH) under very controlled conditions (Brem et al., 2012) and is ignored in our calculations. Thus, we only correct the scattering coefficient to ambient RH in our corrections of SSA. Estimates of the uncertainties in hygroscopic dependence of light scattering coefficient $\sigma$sp are scarce and depend primarily on the uncertainties in RH and in nephelometer-measured scattering coefficient, in addition to system-dependent particle losses in the humidograph. One study ( Titos et al. (2016) ) estimates the uncertainty in hygroscopic $\sigma$sp enhancement for humidographs similar to that deployed at APP and we now propagate this uncertainty through the calculations to estimate uncertainty in SSA.

Changes to Paper:

1. We clarify how the humidograph corrects scattering and backscattering coefficients to ambient RH by adding the following text to Sect. 3.1-Single Scattering Albedo and Scattering Asymmetry Parameter: "The humidograph consists of a humidifier and a second TSI 3563 nephelometer placed downstream of the first nephelometer. A one-hour programmable RH ramp (<40% to 85%) is applied to the air stream entering the second nephelometer. A two-parameter fit of the ratio of humidified to dried aerosol $\sigma$sp is applied to each RH ramp deduce the RH dependence of $\sigma$sp (Eq.3 of Titos et al., 2016). A similar fit is calculated for $\sigma$bsp."

2. We now propagate estimated uncertainties in humidified $\sigma$sp and $\sigma$bsp to estimate uncertainties in RH-corrected SSA and g, for each season. We explain the methodol-

ogy of these corrections in Sect. 3.1 via the following additions:

(a) "Radiative transfer models typically only treat the scattering dependence when correcting $\omega 0$ to ambient RH; and assume that absorption changes negligibly with RH. While this approach may or may not hold true for all aerosol types (ex: some organics, sulfur-coated soot), the dependence of $\sigma$ap on RH is experimentally very difficult for all but laboratory studies (especially at high RH) conducted under very controlled conditions (Brem et al., 2012) and is ignored in our calculations. Thus, we only correct $\sigma$sp to ambient RH in our corrections of $\omega 0$. Uncertainties in correcting $\sigma$sp to ambient RH are due to uncertainties in (1) scattering coefficients measured by the dry and humidified aerosol nephelometers ($\Delta\sigma$sp=9.2%, Supplement to Sherman et al., 2015); and (2) RH inside the humidified nephelometer ($\Delta$RH$\sim$3%; Titos et al., 2016). Titos et al. (2016) used these values as inputs to a Monte Carlo simulation to estimate the uncertainty in the RH-corrected scattering coefficient as $\Delta\sigma$sp$\sim$ 20% (their Fig. 2b) for high-RH (>90%) and for moderately hygroscopic aerosols such as those observed at APP (Sherman et al., 2016b). We apply $\Delta\sigma$sp$\sim$ 20%, along with uncertainty in dried aerosol absorption coefficient ($\Delta\sigma$sp=20%; Sherman et al., 2015), as inputs to Eq. S9 of supplement to Sherman et al. 2015 to calculate $\Delta\omega 0$. Single-scattering albedo uncertainty is larger for more absorbing aerosols and is zero for purely scattering aerosols ($\omega 0$=1). We use monthly median $\omega 0$ values (Fig.5b) to calculate $\Delta\omega 0\sim$0.03 for winter and surrounding months and $\Delta\omega 0\sim$0.02 for summer and surrounding months (Table 2)."

(b) "Uncertainty in the calculated value of g at ambient RH arises due to uncertainties in the measured $\sigma$bsp and $\sigma$sp, each of which is subject to the same measurement uncertainties as outlined above. Sherman et al. (2015) reported a nearly identical uncertainty in dried aerosol hemispheric backscatter coefficient ($\Delta\sigma$bsp=8.9%) as for the scattering coefficient ($\Delta\sigma$sp=9.2%). This, along with the lack of published uncertainties in humidified $\Delta\sigma$bsp for similar experimental configurations as that deployed at APP, lead us to use the same uncertainty estimate for ambient-RH $\Delta\sigma$bsp as for ambient-RH

$\Delta\sigma$sp (∼20%). Inserting the ambient-RH uncertainties $\Delta\sigma$bsp and $\Delta\sigma$sp into Eq.S8 of supplement to Sherman et al. (2015) lead to hemispheric backscatter fraction uncertainty $\Delta$b∼0.0085, which in turn can be used along with the relation between g and b (Eq.3) to calculate $\Delta$g=$|\partial$g/$\partial$b$|$ $\Delta$b ∼0.01."

3. We updated all measurement-based DRE uncertainty values in the manuscript, to reflect the updated measurement uncertainty values $\Delta\omega$0 and $\Delta$g. The new uncertainty estimates do not give rise to any changes in the main results of the paper but we did need to make small modifications to the wording in places of the Results/Discussion and Summary/Conclusion sections (based on these changes)

Reviewer 1 Technical Comment 4: Authors present sensitivity of DRE to surface reflectance (SR) at TOA and surface as 3.3Wm-2 and 2.7Wm-2 during June and 0:22Wm-2 and 0:20Wm-2 during December. What surprises me is the very small difference in SR values at TOA and at surface. It is common knowledge that surface reflectivity will have very little effect on DRE at surface but can have significant effect at TOA. One can read reason for it in Chung (2012). In other words, a significant difference is expected between TOA and surface SR values. See for example Figure 10 of Gadhavi and Jayaraman (2004) who have used similar approach and the same radiative transfer code (SBDART) to calculate DRE (they called it radiative forcing) as a function of AOD and surface reflectance. They have reported that when surface type changes from sea to sand (which is large change in surface reflectance) it causes a little change in DRE at surface but a large change in DRE at TOA for a fixed AOD. The values reported in the current manuscript may not be wrong but a thorough discussion needs to be included why their finding is at variance with others or the common knowledge. I believe such a discussion will add value to their manuscript as it will lead to better understanding of how non-aerosol parameter affects aerosol radiative forcing.

Authors' Response: We agree with the reviewer that more discussion of SR and comparisons of our results with other papers such as Gadhavi and Jayaraman (2004) and McComiskey et al. (2008) could provide insight on the role of surface reflectance in

aerosol DRE. To this end, we have made several related changes to the paper, enumerated below. We partially agree with the reviewer's assertion that "It is common knowledge that surface reflectivity will have very little effect on DRE at surface but can have significant effect at TOA." Surface reflectivity can have either a large or small effect on DRE at TOA and at the surface. The difference between DRE (and DRE sensitivity) at the TOA and that at the surface is (for a fixed AOD) dependent on aerosol absorption and on the relative albedos of the atmosphere and underlying surface, with larger DRE differences (between TOA and surface) for more absorbing aerosols (low SSA) and brighter surfaces and smaller differences for less absorbing aerosols (higher SSA) and darker surfaces. Chung's Fig.5 depicts the case for very dark aerosols (SSA=0.19), which is mainly applicable to local sources of black carbon aerosols. Chung also states (first paragraph of their Sect.3) that "The surface plays an important role in case of absorbing aerosols (i.e., aerosols with low SSA). As Fig. 5 shows, higher albedo (i.e., more reflection at the surface) increases aerosol absorption and thus aerosol forcing at the TOA as well as in the atmosphere. Higher albedo increases aerosol absorption because absorbing aerosols absorb not just the downward solar radiation but also the reflected upward radiation. Higher albedo also decreases aerosol scattering back to the space, further contributing to higher aerosol forcing at TOA. Ice, snow and desert have high surface albedo.)". The aerosols at APP are 'bright' (SSA∼0.91-0.96) and the surface is fairly dark (Fig(s).3 of manuscript) so the effects of changes in DRE due to small changes in surface reflectance (i.e. the DRE sensitivity SR) will be relatively small, both at the TOA and at the surface. Our results are consistent with McComiskey et al. (2008), who also reported small differences (∼10-20%) between SR at the TOA and SR at the surface for a continental US site, a tropical Pacific site, and a site in Alaska (their Fig.4).

Changes to paper:

1. The SR values previously calculated (and referenced above by reviewer) were incorrectly calculated and we thank the reviewer for catching this! We now report sensitivity to surface albedo SR as the slope of plot of DRE versus broadband (spectrally-averaged) R, not relative surface reflectance (as we had mistakenly calculated it). This facilitates more direct comparisons with McComiskey et al (2008) and Gadhavi and Jayaraman (2004), in addition to corrected DRE uncertainties due to the use of the proper SR values. There are no major changes in the main results of the paper but small changes in the DRE uncertainty values and larger absolute SR values, that are in much better agreement with McComiskey et al. (2008).

2. We modified the following passages to Sect. 4.3, more clearly explaining how SR is calculated.

(a) "For the sensitivity SR, we scale the entire spectral surface reflectance curve (Figs.3) by proportionally scaling the input surface type coefficients supplied to SB-DART (Fig.3), to vary the broadband (250-4000nm) surface reflectance R (Figs. 6d and 7d). For example, doubling both the sand and vegetation coefficient values supplied to SBDART scales the entire surface reflectance curve by the same amount, thereby doubling the base-case value of broadband R in Table 1."

(b) "The base case R values in Table 1 are the broadband surface reflectance corresponding to the monthly mean spectral surface reflectance curves (Figs.3). We then vary the independent variables i individually about these base case values (Table 1) to generate the 'seasonal' DRE versus i curves. We evaluate $S_i = \partial$ (DRE) / $\partial i$, at base case i value, as the regression slope of the five points on each side of the base case value

3. We re-made the plots of SR (Figs. 6c and 7c) so that they now plot curves of DRE versus spectrally-averaged R (Change 1 above), not DRE versus relative surface reflectance.

4. We compare our SR values to those of McComiskey et al (2008) and Gadhavi and Jayaraman (2004) and discuss the results in Sect. 5.2, with the paragraph shown below. We also added curves of DRE versus AOD for the same surface types as ,

Gadhavi and Jayaraman (2004), to the supplement of our paper. This allowed for a more direct comparison of our SR results with theirs.

"Relatively low sensitivity of DRE to surface reflectance at APP during the study period is seen by the low SR values, which range from 17 Wm-2 (14 Wm-2) at the TOA (surface) during JUN to 2.0 (1.8) Wm-2 at TOA (surface) during DEC, per unit change in R (Table 3). Surface reflectance changes very little during vegetative summer months and large changes within a given season typically only occur for extended snow cover during some winter months, when SR is very small. Our SR values for SEP (9.1 Wm-2at TOA and 7.7 Wm-2 at surface, per unit change in R) are very close to those reported by McComiskey, et al. (2008) for the SGP site (their Fig.4). While TOA SR values in our study and that of McComiskey et al. (2008) are only ∼10-20% larger than SR values at the surface, Gadhavi and Jayaraman (2004) reported much higher DRE sensitivity to surface type at the TOA than at the surface in Antarctica. As part of their analysis, Gadhavi and Jayaraman (2004) plotted both TOA and surface DRE versus AOD for different surface types (snow, seawater, sand, and vegetation), using the same SB-DART radiative transfer code used in our study and that of McCommiskey et al. (2008). Gadhavi and Jayaraman (2004) reported similar TOA DRE sensitivities to changes in surface albedo as we do. Their TOA DRE changed by ∼10 W m-2 as they changed the surface type from all sea water to all snow (their Fig.10, for AOD=0.1), which represents close to a unit change in surface albedo. However, their corresponding change in surface DRE was only ∼3 W m-2. To determine whether the much smaller surface SR reported by Gadhavi and Jayaraman (2004) is due to the method used to report them, we plot similar curves of DRE versus AOD for each of the four surface types (Fig.S3 of supplement to this paper). The differences of DRE calculated for pure snow and pure seawater (at AOD=0.10) are similar to our SR values derived using the sensitivity curves (Fig.5), both at the TOA and at the surface. This leads us to speculate that the smaller surface SR reported by Gadhavi and Jayaraman (2004) could be due to differences in aerosol properties and/or their spectral dependence supplied to the SBDART model, or else on how spectral surface reflectance was parameterized. Gadhavi and

Jayaraman (2004) did not state the spectral $\omega_0$ used as input to the RTM and aerosol absorption can have a very large effect on DRE sensitivity to the underlying surface type (Chung, 2012)."

Reviewer 1 Technical Comment 5: Page 35 caption of Fig. 1: Longitude number of the site should have suffix "W".

Authors' Response: Thanks for catching that!

Change to Paper: We have corrected the longitude in Fig.1, changing "N" to "W",

Please also note the supplement to this comment:
https://www.atmos-chem-phys-discuss.net/acp-2017-513/acp-2017-513-AC3-supplement.pdf